# Direct functionalization of methane into ethanol over copper modified polymeric carbon nitride via photocatalysis

Yuanyi Zhou[1,2], Ling Zhang[1,3] & Wenzhong Wang [1,3]

Direct valorization of methane to its alcohol derivative remains a great challenge. Photocatalysis arises as a promising green strategy which could exploit hydroxyl radical ($\cdot OH$) to accomplish methane activation. However, both the excessive $\cdot OH$ from direct $H_2O$ oxidation and the neglect of methane activation on the material would cause deep mineralization. Here we introduce Cu species into polymeric carbon nitride (PCN), accomplishing photocatalytic anaerobic methane conversion for the first time with an ethanol productivity of 106 μmol $g_{cat}^{-1} h^{-1}$. Cu modified PCN could manage generation and in situ decomposition of $H_2O_2$ to produce $\cdot OH$, of which Cu species are also active sites for methane adsorption and activation. These features avoid excess $\cdot OH$ for overoxidation and facilitate methane conversion. Moreover, a hypothetic mechanism through a methane-methanol-ethanol pathway is proposed, emphasizing the synergy of Cu species and the adjacent C atom in PCN for obtaining $C_2$ product.

---

[1] State Key Laboratory of High Performance Ceramics and Superfine Microstructure, Shanghai Institute of Ceramics, Chinese Academy of Sciences, 1295 Dingxi Road, Shanghai 200050, People's Republic of China. [2] University of Chinese Academy of Sciences, Beijing 100049, People's Republic of China. [3] Center of Materials Science and Optoelectronics Engineering, University of Chinese Academy of Sciences, Beijing 100049, People's Republic of China. Correspondence and requests for materials should be addressed to W.W. (email: wzwang@mail.sic.ac.cn)

As the principal component of natural gas, methane is an important fossil energy which is prized for its abundance and cleanliness[1,2]. Its oxygenated derivatives, especially alcohol derivatives, are not only transportable and storable liquid fuels, but also versatile building blocks of value-added chemicals[3]. However, methane conversion to its alcohol derivatives is difficult to master and has rightfully emerged as the 'Holy Grail' of catalysis[4]. Methane exhibits perfectly symmetrical tetrahedral structure with high C−H bond strength ($440 \, kJ \, mol^{-1}$) and negligible electron affinity, requiring harsh conditions for its activation[5]. On the other hand, it is also hard to preserve the products from deep mineralization under practical situations since all the derivatives are easier to become activated for oxidation[6]. Such prospects and challenges of methane conversion have captured the interest of both academic community and industry.

So far, due to the inert nature of methane, the current industrial route for methane valorization is indirect and depends on preliminary high-temperature and high-pressure oxidation to syngas, then comes liquid fuels and commodity chemicals[7–9]. Direct upgrading of methane through photocatalysis arises as a promising green strategy, which allows promoting difficult reactions under mild conditions by virtue of photoexcitation instead of thermal activation. Photocatalytic conversion of methane to liquid fuels still remains in its infancy as only few attentions were paid. $WO_3$ has been a classic material in this field supplying hydroxyl radical (·OH) through $H_2O$ oxidation to turn methane into methanol[10–13]. In addition to $WO_3$, bismuth-based compounds such as $BiVO_4$ and $Bi_2WO_6$ have also been estimated[14], while reports of V containing zeolites have emerged recently about tuning the surface acid−base properties to a better methanol selectivity[15,16]. In these researches, the highest methanol production rate is $67.5 \, \mu mol \, g_{cat}^{-1} \, h^{-1}$ from $WO_3$ by using electron scavenger[12], which is still far from satisfaction. Thus, refinement of material design and in-depth understanding of the mechanism are needed to improve the production of liquid fuel.

Among the reported articles, the dominant way to activate methane is oxidizing $H_2O$ directly to produce ·OH, which would abstract a hydrogen atom from methane to generate methyl radical. However, taking into account that the intermediates from methane conversion are highly susceptible to deep oxidation by excess ·OH to $CO_2$[3], an appropriate amount of ·OH and gentle path to produce it should be the key factors for selective partial oxidation of methane. Although the decomposition of $H_2O_2$ by Fenton process is an efficient way to obtain ·OH[17], adding $H_2O_2$ usually could not receive desirable results in photocatalytic selective oxidation or even be helpless because only the catalyst-surface bound species are sufficiently mild to be highly selective[18]. Thus, constructing active sites to manage generation and in situ decomposition of $H_2O_2$ to produce ·OH in the proper amount could be a possible strategy to avoid deep mineralization during methane conversion.

In another sense, methane activation through adsorption is the same critical as in situ production of the reactive oxygen species. The interaction between methane molecule and the surface of photocatalyst would induce subtle change to the perfect tetrahedral symmetry of methane, which would be helpful for selective methane conversion. Bypassing this process upon the surface of photocatalyst but focusing solely on the generation of radicals, becomes one of the reasons that the efficiency of photocatalytic methane conversion is still far from satisfaction. For Cu-exchanged zeolite, a typical thermal catalyst of methane conversion, it is the synergy of Cu species for activation and adjacent lattice oxygen for conversion that brings about considerable methanol production[7,19]. Since free methane molecule could hardly escape from complete mineralization, we believe that constructing active sites to adsorb methane and cooperate with

the reactive oxygen species plays an essential role in photocatalytic methane conversion.

To achieve the objective, polymeric carbon nitride (PCN) is a competent candidate, which has been investigated as one of the potential next-generation materials for energy conversion and catalysis application[20–23]. Its appealing electronic structure could oxidize $H_2O$ into $H_2O_2$[24,25], meanwhile, well-ordered mesoporous cavity and lone-pair electrons originated from the edge N atoms (C−N=C) allow PCN to interact with exogenous matter, making it a perfect molecular scaffold for binding or intercalation of exotic atoms[26,27]. Considering the mild conditions for methane conversion, the active site of enzyme offers excellent reference to search the potential components for PCN modification. In the methanotrophic bacteria, particulate methane monooxygenase is a critical metalloenzyme with a unique Cu cluster as the active site of methane hydroxylation[28].

In the present work, we intend to introduce Cu species into the cavity of PCN inspired by its capability of $H_2O_2$ decomposition and the active site of particulate methane monooxygenase. Generation and in situ decomposition of $H_2O_2$ to produce ·OH are managed over Cu modified PCN, of which Cu species are also active sites for methane adsorption and activation. These features avoid excess ·OH for deep mineralization, facilitating photocatalytic anaerobic methane conversion and generating ethanol as the main liquid product at a rate of $106 \, \mu mol \, g_{cat}^{-1} \, h^{-1}$ under visible light. Moreover, the synergy of Cu species and the adjacent C atom in PCN plays a key role to obtain ethanol through a methane–methanol–ethanol pathway.

## Results

**Structure and morphology.** The samples were prepared by thermal condensation of the precursor, denoted as PCN or Cu-X/PCN, where X corresponds to the theoretical weight percentage of Cu. Chemical structure of PCN and a series of Cu-X/PCN was first characterized by X-ray diffraction (XRD). Cu-X/PCN held similar patterns to that of PCN without additional peaks. However, the intensity ratio of 13.0° (in-plane packing) to 27.4° (interfacial stacking) peaks gradually decreased from 0.37 to 0.18 by incorporating more Cu into PCN, revealing that the interfacial stacking periodicity of PCN had been destructed (Supplementary Fig. 1)[20,21]. In Fourier transform infrared (FTIR) spectra of PCN and Cu-0.5/PCN (Supplementary Fig. 2), no distinct changes of the band positions occurred after Cu introduction, confirming that the modification of PCN exerted a negligible influence on its basic structure. Moreover, no more than the layered structure of PCN was observed from transmission electron microscopy (TEM) images with Cu incorporation (Supplementary Fig. 3).

In terms of the valence state and bonding situation of Cu, the most direct evidence comes from X-ray photoelectron spectroscopy (XPS) analyses. In the Cu 2p XPS spectrum of Cu-0.5/PCN (Fig. 1a), the Cu element is of either $Cu^I$ or $Cu^0$ according to the Cu $2p_{1/2}$ and Cu $2p_{3/2}$ peaks at 952.3 and 932.5 eV, respectively, whereas the signal of $Cu^{II}$ was submerged in the background[23]. No peak of the Cu LMM (Supplementary Fig. 4a) was detected but noises, meaning that $Cu^0$ could not be told from $Cu^I$[29]. Electron spin resonance (ESR) spectrum (Fig. 1b) offered assistance to learn more details about the Cu species. A typical signal of $Cu^{II}$ was received with $g$ values about 2.08[30]. Interestingly, nine small peaks with equal spacing were found by careful inspection, which might signify hyperfine coupling from partial delocalization of the unpaired electron spinning over two Cu centers[31]. These findings support the mixed-valence conjecture. The form of the Cu species was examined through the O 1s XPS spectra (Supplementary Fig. 4b), indicating that mixed-valence Cu species formed a complex with PCN instead of oxides. The next question would be its bonding situation. Both PCN and Cu-0.5/PCN from the C 1s XPS spectra (Supplementary Fig. 4c)

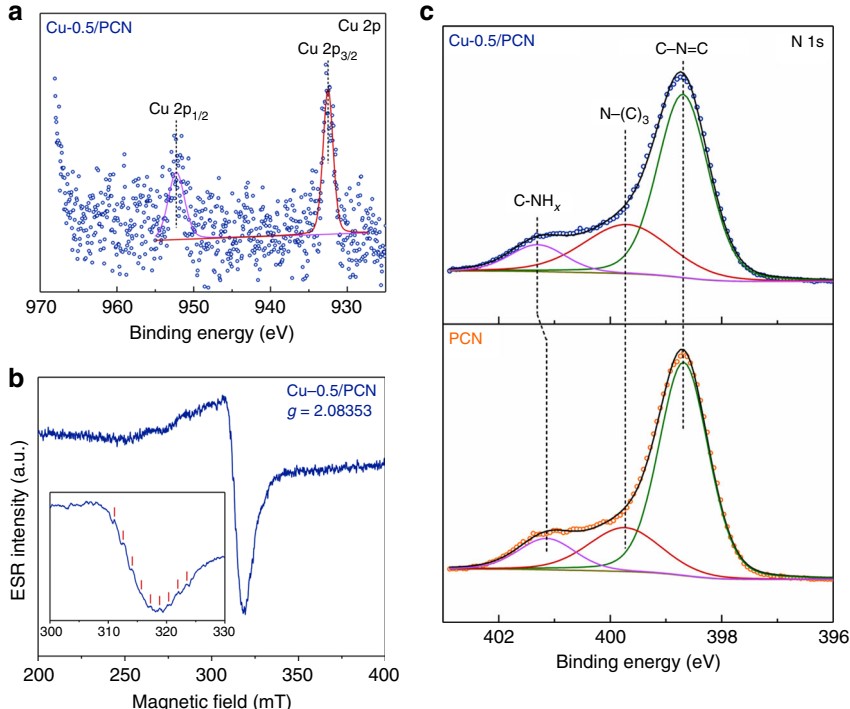

**Fig. 1** Valence state and bonding situation. **a** Cu 2p XPS spectrum of Cu-0.5/PCN. **b** ESR spectrum of Cu-0.5/PCN. **c** N 1s XPS spectra of PCN and Cu-0.5/PCN

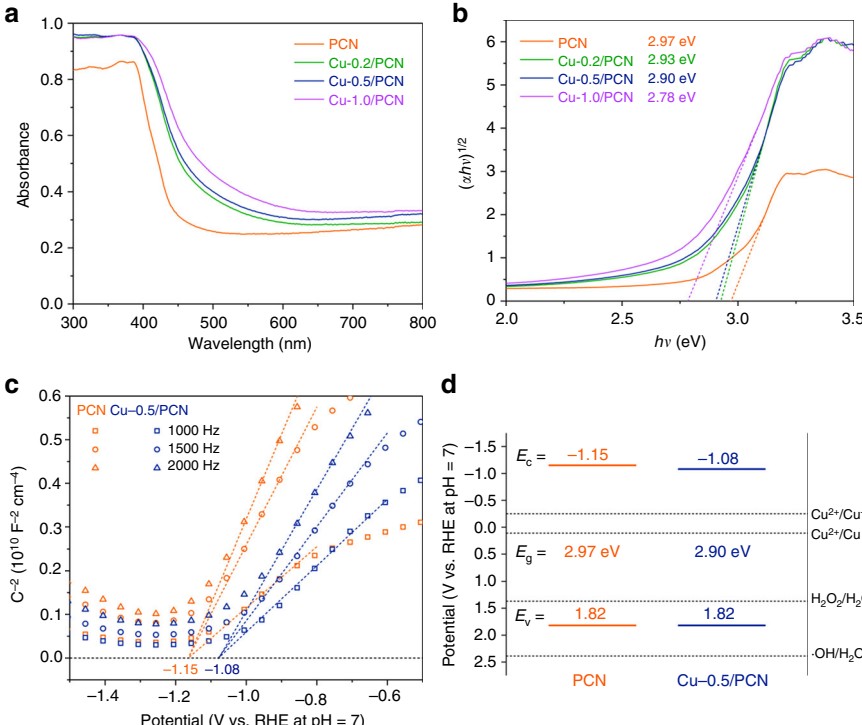

**Fig. 2** Optical and electrochemical characterizations. **a** DRS spectra of PCN and Cu-X/PCN. **b** Plots of transformed Kubelka–Munk function versus photon energy for PCN and Cu-X/PCN. **c** Mott–Schottky plots of PCN and Cu-0.5/PCN. **d** Band structure alignments of PCN and Cu-0.5/PCN

could be deconvoluted into identical components without any shift. In Fig. 1c, the peak in the N 1s XPS spectrum of PCN at 401.1 eV representing hydrogen-bearing amine C−NH$_x$ raised to 401.3 eV after Cu introduction, implying the reduced electron density of the N atoms which was typical for complex formation[32,33]. This result suggested that the mixed-valence Cu species were filled into the heptazine rings and coordinated with the sp$^2$-bonded N atoms. The element content of PCN and Cu-

0.5/PCN was also analyzed (Supplementary Table 1). No residual Cl was detected, which might be carried out in the form of hydrogen chloride during thermal condensation.

**Optical and electrochemical characterization.** Filling Cu species into N conjugated aromatic pores has dramatically altered the optical property of the samples. Progressive redshift at absorption edge was attained with the introduction of more Cu species (Fig. 2a), while bandgaps of the samples determined by Kubelka–Munk function (Fig. 2b) were gradually narrowed from 2.97 to 2.78 eV.

Mott–Schottky plots (Fig. 2c) helped to estimate the approximate conduction band positions, revealing band structure alignments (Fig. 2d) together with the optical bandgap assessed from the data of diffuse reflectance spectroscopy (DRS). Valence band position of 1.82 V implied a weaker oxidizing ability of Cu-0.5/PCN than WO₃, BiVO₄ and so on[14], which located between the oxidation level for $H_2O$ to $H_2O_2$ and to ·OH, indicating that holes in the valence band could oxidize $H_2O$ into $H_2O_2$ through a two-electron pathway rather than ·OH directly[25]. Conduction band of Cu-0.5/PCN was also positioned properly to reduce the Cu species and complete the photocatalytic cycle. These alignments conform to a moderate condition for producing surface bound $H_2O_2$.

**Photocatalytic characterization.** The characterizations gleaned above have supported our anticipated material design, we were then in a position to verify the idea on methane conversion. As PCN was chosen to produce $H_2O_2$ from $H_2O$ and Cu species incorporated to accomplish in situ generation of ·OH, we examined the photocatalytic $H_2O_2$ and ·OH production over PCN and Cu-0.5/PCN without $O_2$ (Fig. 3a, b). A $H_2O_2$ production rate of 12.5 μmol $g_{cat}^{-1}$ $h^{-1}$ was obtained over PCN, whereas it seemed negligible over Cu-0.5/PCN. In contrast, a more intense fluorescent signal over Cu-0.5/PCN implied that Cu modification accelerated the decomposition of $H_2O_2$ and produced more ·OH, while the weak signal over PCN was attributed to the inefficient decomposition by photolysis or photogenerated electron attack $(H_2O_2 + e^- \rightarrow \cdot OH + OH^-)$[17]. This result demonstrated that under illumination Cu-0.5/PCN could oxidize $H_2O$ into surface bound $H_2O_2$ and simultaneously decompose it into ·OH, which might be a moderate oxidation agent.

Another purpose of Cu modification was to facilitate methane activation inspired by the active sites of particulate methane monooxygenase. Adsorption behavior was examined with the help of in situ infrared (IR) technique. After the admission of methane onto Cu-0.5/PCN, two distinct IR bands at 3015 and 1304 cm$^{-1}$ were observed (Fig. 3c), shifting toward lower

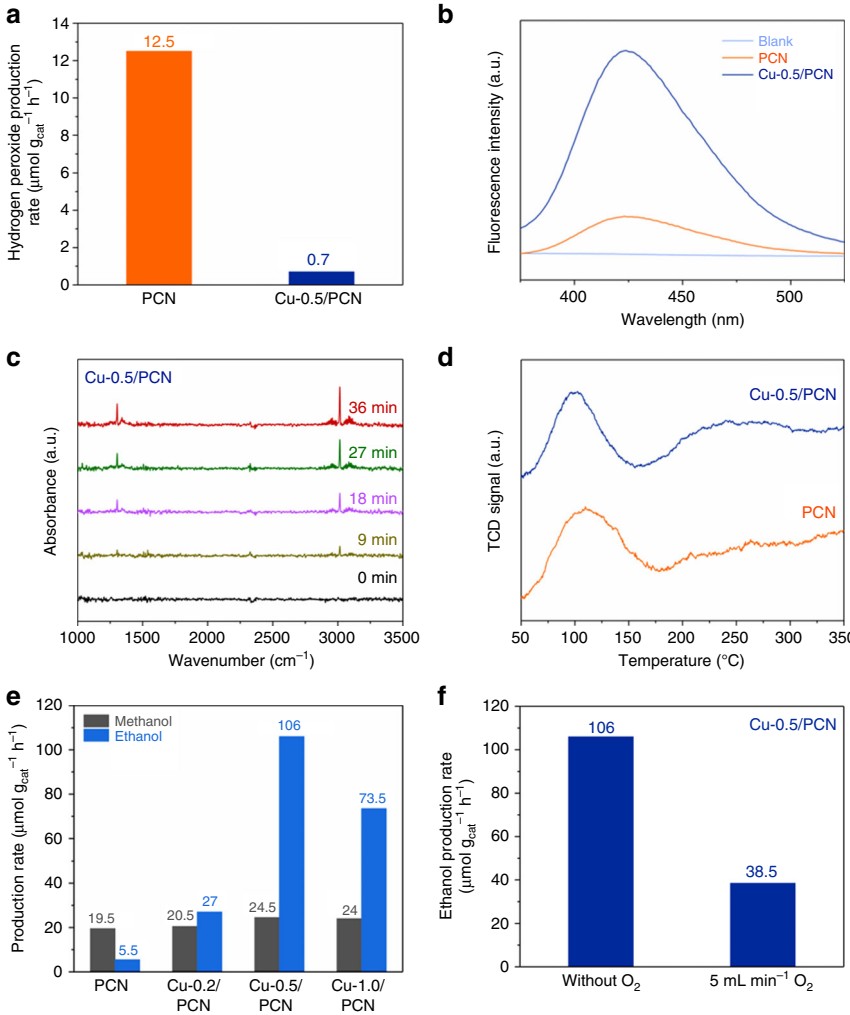

**Fig. 3** Photocatalytic performance. **a** Photocatalytic anaerobic $H_2O_2$ production over PCN and Cu-0.5/PCN. **b** Fluorescent spectra of 2-hydroxyterephthalic acid for hydroxyl radical detection over PCN and Cu-0.5/PCN. **c** In situ IR spectra of methane adsorption on Cu-0.5/PCN. **d** Methane TPD of PCN and Cu-0.5/PCN. **e** Liquid products of methane conversion over PCN and Cu-X/PCN. **f** Photocatalytic methane conversion over Cu-0.5/PCN with or without $O_2$

wavenumbers than that of free methane molecules (3020 and 1306 cm$^{-1}$, respectively) due to methane adsorption[34]. Intensities of these bands on Cu-0.5/PCN increased slowly, while on PCN they kept unchanged (Supplementary Fig. 5), indicating that Cu introduction might benefit methane adsorption and activation. Methane temperature programmed desorption (TPD) of PCN and Cu-0.5/PCN was also investigated to gain more insight into methane adsorption and activation. As shown in Fig. 3d, the physisorption of methane (around 100 °C) on PCN was stronger than Cu-0.5/PCN, agreeing with the unchanged signal of the in situ IR spectra on PCN. After introducing Cu into PCN, the peak around 250 °C demonstrated the chemisorption of methane, agreeing with the progressively increasing signal of the in situ IR spectra on Cu-0.5/PCN[35]. This result, together with the in situ IR spectra, implied that PCN favored methane enrichment and Cu species coordinated into PCN played a key role in the C−H activation. Since the active sites for both in situ generation of ·OH and methane adsorption were adjacent sites, the cooperation of these functions might facilitate methane conversion.

To validate the feasibility of our design, a series of experiments in the liquid–solid dynamic condition for photocatalytic anaerobic methane conversion were performed at room temperature and atmospheric pressure while the accumulated liquid products were analyzed by gas chromatographer. Blank experiments revealed that nothing occurred without photocatalysts or without methane admission, ruling out photochemical reactions as well as carbon source from photocatalyst itself. As depicted in Fig. 3e, the liquid products contained methanol and ethanol. The variation of methanol productivity over all the samples was subtle. However, a significant increase of ethanol yield was achieved over Cu-X/PCN, among which Cu-0.5/PCN reached the highest ethanol production rate of 106 μmol $g_{cat}^{-1}$ h$^{-1}$. The time course of the photocatalytic methane-to-ethanol conversion over Cu-0.5/PCN was carried out (Supplementary Fig. 6), indicating that the ethanol production rate decayed slightly over 24 h of testing. The XPS spectra of the sample after cycling tests were also studied. From the Cu 2p XPS spectra (Supplementary Fig. 7a), the mixed-valence state remained unchanged after photocatalytic tests, implying that the oxidized Cu species from $H_2O_2$ decomposition were reduced by the photogenerated electrons. The peak at 284.5 eV in the C 1s XPS spectra became obvious, corresponding to coke deposition and agreeing with the ethanol production decay on Cu-0.5/PCN (Supplementary Fig. 7b). Furthermore, the photocatalytic gas byproducts of methane conversion over Cu-0.5/PCN containing $H_2$, CO and ethane were detected in the gas–solid static condition (Table 1). Despite the conduction band of Cu-0.5/PCN was suitable for $H_2$ evolution, no $H_2$ was detected in the methane-free experiment (Supplementary Fig. 8). Thus, $H_2$ evolution could be excluded from photocatalytic $H_2O$ splitting. It originated from the radical process of methane conversion. However, the $H_2$ evolution rate was rather low and nonstoichiometric to that of alcohols, we ascribed the disappeared $H_2$ to the strong coordination ability of the N atoms on Cu-0.5/PCN to trap the hydrogen atoms[26,36], which was confirmed in the N 1s XPS spectra (Supplementary Fig. 7c) that the electron density of all kinds of N atoms became

higher in comparison to the fresh photocatalyst. It is also worth noting that introducing $O_2$ into the system would result in ethanol production decay (Fig. 3f). Although $O_2$ promoted the yield of $H_2O_2$ over PCN (Supplementary Fig. 9), it also served as electron scavenger and became superoxide radical, which would not only retard the photocatalytic cycle of the Cu species but also initiate further oxidation of the methyl radicals to HCHO and $CO_2$[37].

## Discussion

Based on the results and discussion above, we proposed a hypothetic radical mechanism for photocatalytic anaerobic methane conversion (Fig. 4). Given that holes in the valence band of Cu-0.5/PCN cannot oxidize $H_2O$ into ·OH directly, the generation of ·OH was conducted through a two-electron pathway to form $H_2O_2$[25], which was then in situ decomposed into ·OH by means of the mixed-valence Cu species (Equations (1)–(3))[38]. Meanwhile, electrons reduced the Cu species to complete the photocatalytic cycle and maintain the mixed-valence states (Equations (4) and (5)). Methyl radicals were formed via hydrogen abstraction by ·OH (Equation (6)). This initiation step activated the adsorbed methane and also consumed some of the generated ·OH, which could effectively avoid complete mineralization by excess ·OH. Subsequently, some of the generated methyl radical underwent radical coupling to produce ethane (Equation (7)) and the other reacted with $H_2O$ to produce methanol (Equation (8)). The ethane would also be attacked by ·OH to produce ethyl radical (Equation (9)), while the following ethanol was acquired from the reaction between ethyl radical and $H_2O$ (Equation (10)).

$$2H_2O + 2h^+ \rightarrow H_2O_2 + 2H^+ \tag{1}$$

$$Cu^0 + H_2O_2 \rightarrow Cu^I + OH^- + \cdot OH \tag{2}$$

$$Cu^I + H_2O_2 \rightarrow Cu^{II} + OH^- + \cdot OH \tag{3}$$

$$Cu^I + e^- \rightarrow Cu^0 \tag{4}$$

$$Cu^{II} + e^- \rightarrow Cu^I \tag{5}$$

$$CH_4 + \cdot OH \rightarrow \cdot CH_3 + H_2O \tag{6}$$

$$\cdot CH_3 + \cdot CH_3 \rightarrow C_2H_6 \tag{7}$$

$$\cdot CH_3 + H_2O \rightarrow CH_3OH \tag{8}$$

$$C_2H_6 + \cdot OH \rightarrow \cdot CH_2CH_3 + H_2O \tag{9}$$

$$\cdot CH_2CH_3 + H_2O \rightarrow C_2H_5OH + \cdot H \tag{10}$$

$$\cdot H + \cdot H \rightarrow H_2 \tag{11}$$

According to the radical mechanism above, the generation of methanol in the liquid–solid dynamic condition should be more efficient like PCN (Fig. 3e) because the content of methane was far above ethane. However, for Cu-X/PCN, the production rate of methanol was inferior to that of ethanol. Given the poor solubility of alkane in $H_2O$, there might be another mechanism dominating methanol conversion into ethanol for Cu modified PCN.

In a previous report, Jiao et al. presented a synergistic effect in Cu-$C_3N_4$ facilitating electrocatalytic $CO_2$ reduction to $C_2$ products[26], of which the Cu species coordinated to the carbonous intermediates while the adjacent C atom in $C_3N_4$ coordinated to the oxygenous ones. Among the mentioned intermediates,

**Table 1 Photocatalytic products of methane conversion over Cu-0.5/PCN[a]**

| Liquid product (μmol $g_{cat}^{-1}$ h$^{-1}$) | | Gas product (μmol $g_{cat}^{-1}$ h$^{-1}$) | | |
|---|---|---|---|---|
| CH$_3$OH | CH$_3$CH$_2$OH | H$_2$ | CO | C$_2$H$_6$ |
| 5.5 | 21.0 | 7.0 | 2.7 | 13.9 |

[a]Gas–solid static condition: 20 mg of photocatalyst strewed in a glass dish surrounded by 25 mL of water, CH$_4$/N$_2$ atmosphere, 500 W Xe-lamp irradiating for 1 h

hydroxymethyl group and methoxy group could be also derived from the interaction between methanol and ·OH[39]. Hence, the dual active center model of the Cu species and the adjacent C atom in PCN might be applicable in our case as well. A series of experiments were carried out to validate the conjecture (Table 2). We started with introducing a small amount of methanol into the system for photocatalytic methane conversion (Entry 2 and 5). A significant increase of ethanol production on both PCN and Cu-0.5/PCN was achieved but more methanol was consumed on PCN than that on Cu-0.5/PCN. Then, experiments with methanol in the absence of methane revealed that more methanol was converted into ethanol with Cu modification, while on PCN it just decomposed, further confirming the role of methanol as a key intermediate (Entry 3 and 6). On the base of the results above, another hypothetic mechanism for methane conversion into ethanol through a methane–methanol–ethanol pathway was proposed (Fig. 4). Hydrogen abstraction of the intermediate methanol by ·OH generated hydroxymethyl and methoxy groups (Equations (12) and (13)). Hydroxymethyl or methyl groups binding on the Cu species coupled with methoxy groups binding on the adjacent C atom in PCN to produce ethanol or ethyl radical, leaving an adsorbed O atom or hydroxyl group, which reacted with hydrogen atoms to form $H_2O$ (Equations (14)–(17)).

$$CH_3OH + \cdot OH \rightarrow *CH_2OH + H_2O \qquad (12)$$

$$CH_3OH + \cdot OH \rightarrow *OCH_3 + H_2O \qquad (13)$$

$$*CH_2OH + *OCH_3 \rightarrow C_2H_5OH + *O \qquad (14)$$

$$*CH_3 + *OCH_3 \rightarrow \cdot CH_2CH_3 + *OH \qquad (15)$$

$$*O + \cdot H + \cdot H \rightarrow H_2O \qquad (16)$$

$$*OH + \cdot H \rightarrow H_2O \qquad (17)$$

That is to say, the core of photocatalytic methane conversion lies in controlling the generation of reactive oxygen species and activating methane. The appealing band structure alignments of PCN holds one key to obtain $H_2O_2$ through $H_2O$ oxidation and reduce Cu species to accomplish photocatalytic cycle, while the mixed-valence Cu species hold another to activate methane and decompose $H_2O_2$ into ·OH. These essentials facilitate methane conversion and largely mitigate the negative effect of excess ·OH, leading to an enhanced efficiency of photocatalytic methane conversion. For producing $C_2$ product, the synergy of the Cu species and the adjacent C atom in PCN provides key contributions. It is also reported that the mixed-valence Cu species were critical in electrocatalytic $CO_2$ reduction to obtain $C_2$ products[40,41]. Thus, in our work, maintaining the mixed-valence

states of Cu species through the photocatalytic cycle might also be a key factor to obtain ethanol.

In summary, systematic experiments revealed that photogenerated holes from PCN could oxidize $H_2O$ into $H_2O_2$ and low valence Cu species could accelerate its in situ decomposition into ·OH which participated in hydrogen abstraction to initiate methane conversion. Meanwhile, photogenerated electrons reduced the Cu species, which were oxidized by $H_2O_2$, to complete the photocatalytic cycle and maintain the mixed-valence states. Cu species coordinated into PCN were also responsible for methane adsorption and activation. This material design provided active sites for in situ generation of ·OH as well as methane adsorption and activation, led to enhanced photocatalytic anaerobic methane conversion and avoided complete mineralization. Furthermore, the synergy between the Cu species and the adjacent C atom in PCN played a key role to obtain $C_2$ product through a methane–methanol–ethanol pathway. Our work put forward a strategy to construct a mild condition for methane activation, broadening the horizon of methane conversion.

## Methods

**Photocatalyst preparation**. All the chemicals involved were of analytical grade and used without further purification. PCN was prepared by thermal condensation of urea. Briefly, 15 g of urea was calcined at 550 °C in muffle furnace for 4 h under

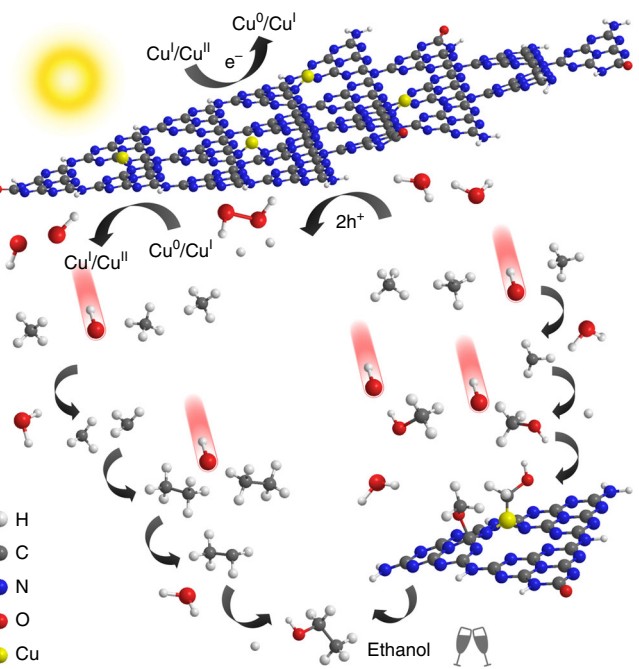

**Fig. 4** The hypothetic mechanism for photocatalytic anaerobic methane conversion over Cu-0.5/PCN

**Table 2 Experiments of methane conversion over PCN and Cu-0.5/PCN for the dual active center model[a]**

| Entry | Catalyst | Medium | Atmosphere | $CH_3OH$ (μmol) | $C_2H_5OH$ (μmol) |
|---|---|---|---|---|---|
| 1 | PCN | $H_2O$ | $CH_4/N_2$ | 0.39 | 0.11 |
| 2 | PCN | 7.5 μmol $CH_3OH$ in $H_2O$ | $CH_4/N_2$ | 2.37 | 0.47 |
| 3 | PCN | 7.5 μmol $CH_3OH$ in $H_2O$ | $N_2$ | 0.82 | 0.40 |
| 4 | Cu-0.5/PCN | $H_2O$ | $CH_4/N_2$ | 0.47 | 2.12 |
| 5 | Cu-0.5/PCN | 7.5 μmol $CH_3OH$ in $H_2O$ | $CH_4/N_2$ | 7.30 | 3.03 |
| 6 | Cu-0.5/PCN | 7.5 μmol $CH_3OH$ in $H_2O$ | $N_2$ | 4.41 | 1.22 |

[a]Liquid–solid dynamic condition: 20 mg of photocatalyst suspended in 25 mL of medium and kept stirring, 100 mL min[−1] of gas flow, 500 W Xe-lamp irradiating for 1 h

air atmosphere at a ramp rate of 10 °C min$^{-1}$. The obtained material (about 0.63 g) was ground into powder and washed sequentially with water several times, then freeze-dried to be used. Cu modified PCN was synthesized as follows. Fifteen gram of urea was dissolved into 30 mL of deionized water and stirred with an aqueous solution of copper chloride (0.1 mol L$^{-1}$, 0.2–1 mL for the samples with different Cu amount), then evaporated at 60 °C overnight. The resulting bluish mixture was heated at a rate of 10 °C min$^{-1}$ to reach 550 °C and maintained for 4 h in air. Subsequent procedure was identical to PCN. Products were denoted as Cu-X/PCN, where X corresponds not only to the volume (mL) of copper chloride solution used, but also to the theoretical weight percentage of Cu fortunately.

**Characterization.** X-ray diffraction (XRD) patterns were collected by a Rigaku Miniflex II desktop X-ray diffractometer with an operating voltage of 30 kV and current of 100 mA, while the wavelength of monochromatized Cu Kα radiation was 0.15418 nm. FTIR spectra were recorded on a Lambda FTIR-7600 spectrometer over 4000–400 cm$^{-1}$ with a resolution of 4 cm$^{-1}$. Morphology and microstructure of the samples were investigated upon a JEOL JEM-2100F TEM under a 200 kV accelerating voltage. XPS measurements were performed at a Thermo Fisher ESCALAB 250Xi XPS microprobe using monochromatic Al Kα radiation (1253.6 eV) as the X-ray source. ESR spectrum was achieved on a JEOL JES-FA200 spectrometer. Diffuse reflectance spectra (DRS) were measured using a Hitachi U-3010 spectrophotometer fitted with an integrating sphere attachment from 300 to 800 nm with barium sulfate as the reference. In situ IR spectra were recorded on a Nicolet iS10 FTIR spectrometer over 3500–1000 cm$^{-1}$. Methane TPD was performed on an AutoChem II 2920 chemisorption analyzer.

**Electrochemical analysis.** Electrochemical analyses were conducted on a Chenhua CHI 660D electrochemical workstation with conventional three-electrode quartz cell system: a platinum sheet and a saturated calomel electrode (SCE) were used as the counter and reference electrodes, respectively. The as-prepared photocatalysts were coated on fluorine-doped tin oxide (FTO) substrates and functioned as working electrodes. Typically, a slurry containing 5 mg photocatalysts and 1.0 mL ethanol was made and ultrasonically scattered for several minutes, then 100 μL of the slurry above was spread onto FTO glass. After natural drying, the working electrodes were then calcined at 120 °C for 2 h to improve the attachment. The electrolyte for the analysis was aqueous 0.1 mol L$^{-1}$ sodium sulfate.

**Hydrogen peroxide detection.** For all the photocatalytic experiments, a customized photochemical reactor was used with a 500 W Xe-lamp (60 mm of spot diameter). The amount of hydrogen peroxide was determined by the potassium iodide spectrophotometric method. Briefly, 20 mg of photocatalyst was dispersed in 25 mL deionized water and kept stirring. Nitrogen (100 mL min$^{-1}$) was bubbled continuously through the suspension in the dark for 30 min, after which the illumination was turned on for 1 h. Subsequently, 1 mL filtrate from the suspension was added into chromogenic reagent containing 4 mL of 0.1 mol L$^{-1}$ potassium iodide and 0.1 mL of 0.01 mol L$^{-1}$ ammonium molybdate. The absorbance at 350 nm after 15 min standing was detected on Hitachi U-3010 spectrophotometer.

**Hydroxyl radical detection.** The formation of hydroxyl radicals was monitored with terephthalic acid as a probe, which could readily capture the radical to produce fluorescent 2-hydroxyterephthalic acid. Typically, 20 mg of photocatalyst was dispersed in 25 mL of 0.5 mmol L$^{-1}$ terephthalic acid dissolved into 2 mmol L$^{-1}$ sodium hydroxide and kept stirring, then the reactor was evacuated by nitrogen and sealed. Fluorescence spectra of 2-hydroxyterephthalic acid after 1 h illumination were measured by Hitachi F-4600 spectrophotometer excited at 315 nm.

**Photocatalytic methane conversion tests.** Photocatalytic anaerobic oxidation of methane involved a series of liquid–solid dynamic experiments. During each test, a suspension of deionized water (25 mL) with the corresponding amount of photocatalyst (20 mg) was added to the reactor and kept in suspension by mechanical stirring. A mixture of methane (10 mL min$^{-1}$) and nitrogen (90 mL min$^{-1}$) was bubbled continuously through the suspension in the dark for 30 min, after which the illumination was turned on for 1 h. After that, filtrate from the suspension was injected to a Techcomp GC7900 gas chromatographer equipped with a SE-54 column and flame ionization detector, to analyze the liquid composition.

**Photocatalytic hydrogen production tests.** To investigate the photocatalytic activity for hydrogen evolution, gas–solid static experiments using the same setup were conducted in consideration of poor solubility of methane in water. In short, 20 mg of photocatalyst was strewed in a glass dish, which was placed in a water bath of 25 mL deionized water, then the reactor was evacuated by a calibrated gas mixture of 10% methane in nitrogen and sealed with rubber septum. After illumination for 2 h, the amount of hydrogen was determined by a Techcomp GC7890II gas chromatographer with a Molecular Sieve 5A 80/100 Mesh column and a thermal conductivity detector. Other gas phase products and the liquid composition were analyzed by a Techcomp GC7900 gas chromatographer with TDX-01, TM-Al$_2$O$_3$/S and SE-54 columns, and flame ionization detectors.

## Data availability

The data that support the findings of this study are available from the corresponding author upon reasonable request.

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

## Acknowledgements

This work was financially supported by the National Natural Science Foundation of China (51772312, 51472260). The authors would like to thank Chen Yan from Shiyanjia Lab (www.shiyanjia.com) for the TPD analysis.

## Author contributions

W.W. and L.Z. conceived the idea and supervised the whole project. Y.Z. designed and carried out the experiments. All the authors discussed the results, contributed to writing the manuscript, commented on the manuscript, and approved the final version of the manuscript for submission.

## Additional information

**Competing interests:** The authors declare no competing interests.

