## [Peer Review File · Nature Communications]

Reviewers' comments:

Reviewer #1 (Remarks to the Author):

This manuscript shows a study of a photocatalytic system with Cu modified C₃N₄ for partial oxidation of methane by water to produce ethanol. However, there are many unclear points as listed below. Thus, this manuscript cannot be recommended for the publication in this journal. The comments are following:

1. As for Supplementary Figures 1 and 2, the reviewer cannot agree with the authors' comments that the XRD patterns and the IR spectrum of the Cu-modified sample(s) are identical to the bare C₃N₄.
2. In Fig. 3C, the signal intensity became large on Cu/C₃N₄ while in Suppl. Fig. 5 methane was strongly adsorbed on C₃N₄ from the initial stage. This means that the bare C₃N₄ is a better absorber.
3. Show the amount of both the edge N atoms and the disappeared hydrogen molecules to clarify the idea of the authors for the explanation of the less amount of hydrogen production rate.
4. In the chemical equations (2) and (3), the meaning of Cu/Cu is unclear. Is it for a pair of Cu atoms? If so, charge balance of the equation is not correct.
5. The proposed mechanism is too speculative. Enough evidences are not shown in this manuscript for example the methyl radical and ethyl radical formation.
6. The proposed mechanism is not reasonable. If this mechanism is correct, they must detect the ethane and methanol also. The coupling methane would provide not ethyl radical but ethane.
7. Why can this photocatalytic system produce ethanol so selectively?
8. The time course of the photocatalytic reaction for long time such as 24 hours is required.
9. The authors must show the material balance and the consumed electron and hole balance (the balance of the reductive products and oxidative products).
10. The photoexcitation mechanism is unclear. What is the photocatalytic sites? Show the wavelength dependence.
11. The authors must show the details for the illumination condition (lamp, intensity, irradiation area, and so on).
12. Show the evidence of ethanol formation, i.e., the gas chromatograms.
13. Some English is difficult to understand.

Reviewer #2 (Remarks to the Author):

As stated by the authors, methane-ethanol conversion is an important goal, and photocatalysis by catalytically active metal nanoparticles supported on carbon nitride materials is an attractive materials combination. The results appear to be quite promising, but not quite enough detail is given to fully evaluate the results. Far too many of the statements in the manuscript are conjecture or qualitative in nature. That is particularly worrying because this has been a very active research topic over the past 10-15 years, and several key recent papers have not been consulted or cited in the bibliography.

First, the authors should note that there was (and can be) only a single "Holy Grail", not several as implied in the Introduction.

Next, the designation of the carbon nitride photoactive catalyst support as "g-C₃N₄" is problematic. Materials prepared by thermolysis from N-rich organic precursors are typically incompletely condensed C_xN_yH_z polymers containing structural units similar to Liebig's melon, with a limiting composition near C₂N₃H. It is extremely difficult to distinguish between s-triazine and heptazine-based structures. The important question of characterizing the chemical composition and structure of C_xN_y and C_xN_yH_z materials has been discussed fully in two recent reviews: Kessler et al, *Nat Rev Mater* 2 (2017) 17030; Miller et al, *Phys Chem Chem Phys* 19 (2017) 15613. Neither of these papers is mentioned here. These should both be consulted and cited in relation to the materials studied here.

In addition, photocatalysis by "graphitic carbon nitride" samples prepared from urea has already been reported by Martin et al, *Angew Chem Int Ed* 53 (2014) 9240. That publication reports a thorough examination of the chemical composition and structure of the material produced, as well as evaluation of the electronic structure from advanced ab initio calculations. Once more, this work is missing from the reference list.

Another key paper that is missing is the work by Schwarz et al (*Adv Mater*, 2017, 1703399) on mixed 2D/3D carbon nitride nanostructures grown from Cu-based solution, and containing Cu nanoparticles. Here also the Cu nanoparticles are characterised by XPS, and the same conclusion is reached - that the particles are either Cu(0) or Cu(I) in nature. The H₂ evolution by photo/electrocatalysis of these heterostructures is evaluated in that paper.

The manuscript can not be accepted in its current form. The authors need to complete their understanding of their materials and photocatalysis results and pay proper attention to previously published work, and use that to construct a more detailed and specific evaluation of their data and the likely mechanisms involved. Those are critical aspects that will be important for future synthesis and processing of materials for large-scale applications.

Reviewer #3 (Remarks to the Author):

In this work, the authors report the design of Cu/g-C₃N₄ materials with tuned properties, for the partial photocatalytic oxidation of methane to ethanol. In fact,

this is a very interesting and rather elusive topic, as these kinds of selective reactions under photocatalytic approaches tend to be difficult, considering the implicit need to balance the oxidizing species in order to avoid total oxidation to CO₂. In this sense, the proposed approach is quite promising, the analysis and experiment design are properly directed, and this approach seems to successfully overcome some of the intrinsic limitations of this reaction, so that it might constitute a reference for future concepts into selective photo-oxidation of methane. Despite this, some aspects should be better addressed and some points must be clarified.

-Please re-write sentence in lines 45-46, page 3: "which is rather low than no matter improvement...", as it doesn't seem very clear.

-Please explain lines 59-60 in the same page.

-Methane monooxygenase is first mentioned in the Introduction. I understand this is a particular enzymatic example that inspired the design of the catalytic structure proposed. It should be more clearly stated as it seems odd in the way it is included.

-In XRD, Figure S1, although the background is relatively noisy, there are some zones in which broad signals seem to be present, such as at 23° and 34° 2theta, for the Cu-0.5/g-C₃N₄. Do these signals correspond to any other phase?

-Regarding g-C₃N₄ characterization, a more in-deep analysis should be performed, considering that urea thermal condensation for obtaining carbon nitrides, normally leads to incomplete polycondensation processes. Is there any remaining unreacted precursor or any influence of the CuCl₂ on the final structure?

-Was Cl analysed or detected by XPS?

-Regarding the Mott-Schottky measurements, it is true that the band structure proposed seems to more or less agree with the expected band positions for the g-C₃N₄. However, it is important to consider that Mott-Schottky conditions might not be fully satisfied: for instance, M-S theory applies to flat surfaces (non-porous) and should be measured under high enough frequencies. In fact, there is a certain variation in the slopes with the frequency, despite the flat band potential seems to be independent from the frequency. I would suggest, at least, not to state "exact conduction band positions", but to indicate it is just an approximation.

-In line 144, what do the authors mean by "photogenerated electron attack"?

-The results suggest that H₂O₂ and further ·OH generation on the Cu-0.5/g-C₃N₄ because of the presence of Cu. This effect might be interesting. Were any tests with the intermediate Cu compositions carried out, in order to see the intrinsic variation of both H₂O₂ and hydroxyl radicals, so that it can be properly correlated to the Cu content?

-In line 155-156 the authors mention the active sites for methane adsorption and activation. Both seem to be related to Cu species. Are these active sites associated to the same form of Cu?

-After the IR measurements in gas-phase, the authors proposed an enhanced CH₄ adsorption on Cu sites. How does the balance between surface ·OH species and

adsorbed CH₄ take place, if both are favored on Cu sites?

-Regarding the blank mentioned tests, please detail how these blank tests were carried out: in absence of catalyst with methane; with catalyst in absence of methane; dark condition tests?

-Was any other possible product analyzed and/or obtained, such as CH₃OH?

-In the experimental part, please indicate characteristic such as purity of the reactants.

- Please indicate if the thermal condensation of the g-C₃N₄ was carried out under air or inert atmosphere.

- Please explain better lines 236-237, as the volume of CuCl₂ solution does not seem to agree with the weight percentage of Cu/urea.

- Please provide details on how the CH₄ adsorption tests were carried out.

- Please state the irradiance used during the photocatalytic tests.

-Regarding the photocatalytic tests: the CH₄/N₂ mixture was continuously supplied to the system during all the test, but the ethanol analyzed from the liquid fraction corresponds to the accumulated alcohol produced. Is it correct? Was the gas phase analyzed in order to assess the possible generation of by-products or CO₂? Was the temperature constant during the test? Were the tests carried out several times for validating reproducibility?

-In the H₂ test, was the CH₄/N₂ mixture initially used for saturating the suspension but the gas flow was stopped before the photocatalytic test?

-As an additional comment, in some sections of the work, the authors mention a "synergistic" effect. Albeit I understand it is associated to a better effect than the sum of the single elements (i.e., g-C₃N₄ and Cu), I would suggest to change this word, as the individual effect was not quantified in order to assess it is synergistic.

Appendix 1

Reviewer #1

This manuscript shows a study of a photocatalytic system with Cu modified C_3N_4 for partial oxidation of methane by water to produce ethanol. However, there are many unclear points as listed below. Thus, this manuscript cannot be recommended for the publication in this journal. The comments are following:

1. As for Supplementary Figures 1 and 2, the reviewer cannot agree with the authors' comments that the XRD patterns and the IR spectrum of the Cu-modified sample(s) are identical to the bare C_3N_4 .

Response

Thanks for the suggestion. Careful inspection of XRD and IR analyses was conducted.

In the XRD patterns, the peak positions kept unchanged with Cu modification. No additional peaks were found owing to the small amount of Cu. However, the intensity ratio of 13.0° (in-plane packing) to 27.4° (interfacial stacking) peaks gradually decreased from 0.37 to 0.18 by incorporating more Cu into g- C_3N_4 , revealing that the interfacial stacking periodicity of g- C_3N_4 had been destructed.

In the IR spectra, no distinct changes of the band positions occurred after Cu introduction. Considering the IR spectra that we have recorded were qualitative rather than quantitative, the transmission difference of the bands between the two samples made no sense. Thus, we came to a cautious conclusion that the modification of g- C_3N_4 exerted a negligible influence on its basic structure.

We have revised relative text in order to avoid misleading expressions. Detailed revision please refer to **Appendix 2-10, 32, 33**.

2. In Fig. 3c, the signal intensity became large on Cu/ C_3N_4 while in Suppl. Fig. 5 methane was strongly adsorbed on C_3N_4 from the initial stage. This means that the bare C_3N_4 is a better absorber.

Response

Thanks for the suggestion. It is an insightful perspective to consider methane adsorption on the materials. We have checked out the results and experimental details of the *in situ* IR

characterization carefully. More evidences that might provide deeper understanding of methane adsorption were also supplemented.

In the *in situ* IR spectra, the adsorption signal occurred on g-C₃N₄ and Cu-0.5/g-C₃N₄ after methane admission, indicating that both of them could adsorb methane. However, the signal intensity became large on Cu-0.5/g-C₃N₄ while it was already intense from the initial stage and remained unchanged on g-C₃N₄, which could be ascribed to different adsorption behaviors. During photocatalysis, chemisorption of methane was more popular as it favored C–H activation. To gain more insight about the influence of Cu introduction on the adsorption behavior, methane temperature programmed desorption (TPD) of g-C₃N₄ and Cu-0.5/g-C₃N₄ were conducted. As shown in Fig. R1, the physisorption of methane (around 100 °C) on g-C₃N₄ was stronger than Cu-0.5/g-C₃N₄, agreeing with the unchanged intense signal of the *in situ* IR spectra on g-C₃N₄. After introducing Cu into g-C₃N₄, the peak around 250 °C demonstrated the chemisorption of methane, agreeing with the progressively increasing signal of the *in situ* IR spectra on Cu-0.5/g-C₃N₄. The results implied that g-C₃N₄ was a good adsorbent and benefit methane enrichment, while introducing Cu species would further improve methane activation.

We have revised the relative description. Detailed revision please refer to **Appendix 2-14**, 29.

Fig. R1. Methane TPD of g-C₃N₄ and Cu-0.5/g-C₃N₄.

3. Show the amount of both the edge N atoms and the disappeared hydrogen molecules to clarify the idea of the authors for the explanation of the less amount of hydrogen production rate.

Response

Thanks for the suggestion. Generally, the content of specified N atom (C–NH_x, N–(C)₃, and C–N=C in our case) was obtained through deconvoluting the N 1s XPS spectrum and calculating

the corresponding integral area. However, from our perspective, the results from this method was subjective rather than objective. So it is not suitable to consult a fuzzy number.

Considering the strong coordination ability of the edge N atoms, we speculated that the disappeared hydrogen atoms were trapped by N atoms. To verify the conjecture, we studied the N 1s XPS spectrum of Cu-0.5/g-C₃N₄ after 24 h tests (Fig. R2). The binding energy of all the peaks became smaller comparing with the fresh photocatalyst, which implied the increasing of the electron density of all kinds of N atoms. This result indicated that after 24 h tests, the disappeared hydrogen atoms were not only trapped by the edge N atoms but also trapped by the sp³-bonded N atoms (C-NH_x and N-(C)₃).

Fig. R2. N 1s XPS spectra of Cu-0.5/g-C₃N₄ before and after 24 h tests.

We have revised the corresponding description. Detailed revision please refer to **Appendix 2-17, 35**.

4. In the chemical equations (2) and (3), the meaning of Cu/Cu is unclear. Is it for a pair of Cu atoms? If so, charge balance of the equation is not correct.

Response

Thanks for the suggestion. Cu⁰/Cu^I meant Cu⁰ or Cu^I, and likewise Cu^I/Cu^{II} meant Cu^I or Cu^{II}. We have corrected the corresponding equations. Detailed revision please refer to **Appendix 2-19**.

5. *The proposed mechanism is too speculative. Enough evidences are not shown in this manuscript for example the methyl radical and ethyl radical formation.*

6. *The proposed mechanism is not reasonable. If this mechanism is correct, they must detect the ethane and methanol also. The coupling methane would provide not ethyl radical but ethane.*

7. *Why can this photocatalytic system produce ethanol so selectively?*

Response for comment 5, 6 and 7

Thanks for the valuable suggestions. More evidences for understanding the photocatalytic methane conversion and the possible mechanism were supplemented.

We attempted to quantify the photocatalytic products of methane conversion over Cu-0.5/g-C₃N₄ in the gas-solid static condition by offline detection. As shown in Table R1, liquid products contained methanol and ethanol while gas products contained H₂, CO and ethane. Thus, as questioned, ethane and methanol besides ethanol were detected.

It has been reported (*Hameed et al., Appl. Catal. A 470 (2014) 327; Villa et al., Catal. Commun. 58 (2015) 200; Agarwal et al., Science 358 (2017) 223 etc.*) that methyl and ethyl radical could be obtained from corresponding light alkanes through hydrogen abstraction by ·OH. With methane input and ethane as byproduct, the generation of methyl and ethyl radicals became possible and the mechanism was further refined as follows. The generation of ·OH and the cycle of Cu species (Eq. 1 to Eq. 5) were still valid. After hydrogen abstraction of methane by ·OH (Eq. 6), some of the generated methyl radical underwent radical coupling to produce ethane (Eq. 7) and the other reacted with water to produce methanol (Eq. 8). The generated ethane would also be attacked by ·OH to produce ethyl radical (Eq. 9), and the following ethanol was acquired from the reaction between ethyl radical and water (Eq. 10).

(9)

Table R1. Photocatalytic products of methane conversion over Cu-0.5/g-C₃N₄^a.

Liquid Product ($\mu\text{mol g}_{\text{cat}}^{-1} \text{h}^{-1}$)		Gas Product ($\mu\text{mol g}_{\text{cat}}^{-1} \text{h}^{-1}$)		
CH ₃ OH	CH ₃ CH ₂ OH	H ₂	CO	C ₂ H ₆
5.5	21.0	7.0	2.7	13.9

^a Gas-solid static condition: 20 mg of photocatalyst strewed in a glass dish surrounded by 25 mL of water, CH₄/N₂ atmosphere, 500 W Xe-lamp irradiating for 1h.

According to the radical mechanism, the generation of methanol in the liquid-solid dynamic condition should be more efficient like g-C₃N₄ because the content of methane was far above ethane. However, for Cu-X/g-C₃N₄, the production rate of methanol was inferior to that of ethanol (Fig. R3). Given the poor solubility of alkane in water, there might be another mechanism dominating methanol conversion into ethanol for Cu modified g-C₃N₄.

Fig. R3. Liquid products of methane conversion over g-C₃N₄ and Cu-X/g-C₃N₄.

Synergistic active centers in Cu-C₃N₄ facilitating electrocatalytic CO₂ reduction to C₂ products was reported recently (Jiao *et al.*, *J. Am. Chem. Soc.* 139 (2017) 18093), of which Cu species coordinated to the carbonous intermediates while the adjacent C atom in C₃N₄ coordinated to the oxygenous ones. Among the mentioned intermediates, hydroxymethyl group and methoxy group could be also derived from the interaction between methanol and $\cdot\text{OH}$. Hence, the dual active

center model of Cu species and the adjacent C atom in g-C₃N₄ might be applicable in our case as well. A series of experiments were carried out to validate the conjecture (Table R2). We started with introducing a small amount of methanol into the system for photocatalytic methane conversion (Entry 2 and 5). A significant increase of ethanol production on both g-C₃N₄ and Cu-0.5/g-C₃N₄ was achieved, but more methanol was consumed on g-C₃N₄ than that on Cu-0.5/g-C₃N₄. Next, the experiments with methanol in the absence of methane revealed that more methanol was converted into ethanol with Cu modification, while on g-C₃N₄ it just decomposed, further confirming the role of methanol as a key intermediate (Entry 3 and 6). On the base of the results above, another hypothetic mechanism for methane conversion into ethanol through a methane-methanol-ethanol pathway was proposed. Hydrogen abstraction of the intermediate methanol by ·OH generated hydroxymethyl and methoxy groups (Eq. 12 and Eq. 13). Hydroxymethyl or methyl groups binding on Cu species coupled with methoxy groups binding on the adjacent C atom in g-C₃N₄ to produce ethanol or ethyl radical, leaving an adsorbed O atom or hydroxyl group, which reacted with hydrogen atoms to form H₂O (Eq. 14 to 17). It was the synergy of the dual active center that brought about an improved ethanol production over Cu modified g-C₃N₄.

Table R2. Experiments of methane conversion over g-C₃N₄ and Cu-0.5/g-C₃N₄ for the dual active center model^a.

Entry	Catalyst	Medium	Atmosphere	CH ₃ OH (μmol)	C ₂ H ₅ OH (μmol)
1	g-C ₃ N ₄	H ₂ O	CH ₄ /N ₂	0.39	0.11
2	g-C ₃ N ₄	7.5 μmol CH ₃ OH in H ₂ O	CH ₄ /N ₂	2.37	0.47
3	g-C ₃ N ₄	7.5 μmol CH ₃ OH in H ₂ O	N ₂	0.82	0.40
4	Cu-0.5/g-C ₃ N ₄	H ₂ O	CH ₄ /N ₂	0.47	2.12
5	Cu-0.5/g-C ₃ N ₄	7.5 μmol CH ₃ OH in H ₂ O	CH ₄ /N ₂	7.30	3.03
6	Cu-0.5/g-C ₃ N ₄	7.5 μmol CH ₃ OH in H ₂ O	N ₂	4.41	1.22

^a Liquid-solid dynamic condition: 20 mg of photocatalyst suspended in 25 mL of medium and kept stirring, 100 mL min⁻¹ of gas flow, 500 W Xe-lamp irradiating for 1 h.

Detailed revision please refer to **Appendix 2-16, 18, 19, 20, 28, 29, 30, 31.**

8. *The time course of the photocatalytic reaction for long time such as 24 hours is required.*

Response

Thanks for the suggestion. The photocatalytic reaction of methane over Cu-0.5/g-C₃N₄ was carried out for 24 h (Fig. R4). Ethanol production decayed slightly during the cycling tests. We studied the XPS spectra of the used sample and ascribed the decay to coke deposition, which corresponded to the peak at 284.5 eV in the C 1s XPS spectrum (Fig. R5). To be cautious, we have rephrased the description about the cycling tests. Detailed revision please refer to **Appendix 2-16, 34, 35.**

Fig. R4. Time course of photocatalytic methane-to-ethanol conversion over Cu-0.5/g-C₃N₄ for 24 h.

Fig. R5. C 1s XPS spectra of Cu-0.5/g-C₃N₄ before and after 24 h tests.

9. The authors must show the material balance and the consumed electron and hole balance (the balance of the reductive products and oxidative products).

Response

Thanks for the suggestion. The photocatalytic methane conversion tests that we performed were in the dynamic condition, while the liquid products were analyzed after irradiation. To analyze the gas byproducts and have insights into the important balances, we tried to quantify the photocatalytic products of methane conversion over Cu-0.5/g-C₃N₄ in the gas-solid static condition by offline detection. As shown in Table R1, liquid products contained methanol and ethanol while gas products contained H₂, CO and ethane. However, because the gas mixture of 10% methane in nitrogen was used for saturating the reactor, we failed to detect the consumed methane as it exceeded the upper limit of the gas chromatograph. It was also hard to quantify the hydrogen atoms trapped on N atoms as well as coke deposition from the XPS spectra. Thus, unfortunately, the material balance and the consumed electron and hole balance could not be evaluated.

10. The photoexcitation mechanism is unclear. What is the photocatalytic sites? Show the wavelength dependence.

Response

Thanks for the suggestion. From the Mott-Schottky plots and the DRS data, we learned the approximate band structure alignments of Cu-0.5/g-C₃N₄, whose valence band position was located between the oxidation level for H₂O to H₂O₂ and to ·OH, while conduction band was positioned properly to reduce Cu species. The results of H₂O₂ and ·OH production confirmed the valence band position. To further verify the conduction band position, Cu 2p XPS spectrum of Cu-0.5/g-C₃N₄ after cycling tests for 24 h was recorded (Fig. R6). The mixed-valence state remained unchanged after photocatalytic tests, indicating that the oxidized Cu species from H₂O₂ decomposition were reduced by the photo-induced electrons. Except the appropriate band structure alignments for photoexcitation, the improved methane adsorption and activation (Fig.R1) as well as methanol conversion into ethanol (Table R2) implied that, the active units constructed by the coordinated Cu species and the adjacent C atom in the g-C₃N₄ framework were the photocatalytic sites. We have complemented the Cu 2p XPS spectra to prove the photoexcitation mechanism. Detailed revision please refer to **Appendix 2-16, 35**.

Fig. R6. Cu 2p XPS spectra of Cu-0.5/g-C₃N₄ before and after 24 h tests.

It is an insightful perspective to consider the wavelength dependence, which would be helpful to evaluate the energy converting efficiency. However, the energy of monochromatic irradiation was so low that the productivity of liquid products remained undetectable in the gas chromatograph. In order to have insight into this important issue, we stepped back to conduct photocatalytic methane conversion using a 420 nm cutoff filter to block out the ultraviolet irradiation (Fig. R7). The ethanol production rate of Cu-0.5/g-C₃N₄ decreased to 29 $\mu\text{mol g}_{\text{cat}}^{-1} \text{h}^{-1}$ from 106 $\mu\text{mol g}_{\text{cat}}^{-1} \text{h}^{-1}$ (full spectrum irradiation), indicating that the major contribution of ethanol production was ultraviolet irradiation.

Fig. R7. Photocatalytic methane conversion over Cu-0.5/g-C₃N₄ in full spectrum and with a 420 nm cutoff filter.

11. *The authors must show the details for the illumination condition (lamp, intensity, irradiation area, and so on).*

Response

Thanks for the suggestion. All the photocatalytic experiments were conducted using a 500 W Xe-lamp with an irradiation area of about 28 cm² (60 mm of spot diameter). We have complemented the illumination condition in the manuscript. Detailed revision please refer to **Appendix 2-26**.

12. *Show the evidence of ethanol formation, i.e., the gas chromatograms.*

Response

Thanks for the suggestion. The gas chromatographic result of the liquid products from photocatalytic methane conversion over Cu-0.5/g-C₃N₄ is displayed as follows (Fig. R8). The peak positions of methanol and ethanol are 1.50 and 1.58 min respectively.

Fig. R8. The gas chromatographic result of the liquid products from photocatalytic methane conversion over Cu-0.5/g-C₃N₄.

13. Some English is difficult to understand.

Response

Thanks for the suggestion. We have checked the manuscript again and corrected the improper or false description. Detailed revision please refer to **Appendix 2-4, 5, 6** and so on.

Reviewer #2

As stated by the authors, methane-ethanol conversion is an important goal, and photocatalysis by catalytically active metal nanoparticles supported on carbon nitride materials is an attractive materials combination. The results appear to be quite promising, but not quite enough detail is given to fully evaluate the results. Far too many of the statements in the manuscript are conjecture or qualitative in nature. That is particularly worrying because this has been a very active research topic over the past 10-15 years, and several key recent papers have not been consulted or cited in the bibliography.

First, the authors should note that there was (and can be) only a single 'Holy Grail', not several as implied in the Introduction.

Next, the designation of the carbon nitride photoactive catalyst support as 'g-C₃N₄' is problematic. Materials prepared by thermolysis from N-rich organic precursors are typically incompletely condensed C_xN_yH_z polymers containing structural units similar to Liebig's melon, with a limiting composition near C₂N₃H. It is extremely difficult to distinguish between s-triazine and heptazine-based structures. The important question of characterizing the chemical composition and structure of C_xN_y and C_xN_yH_z materials has been discussed fully in two recent reviews: Kessler et al, Nat Rev Mater 2 (2017) 17030; Miller et al, Phys Chem Chem Phys 19 (2017) 15613. Neither of these papers is mentioned here. These should both be consulted and cited in relation to the materials studied here.

In addition, photocatalysis by 'graphitic carbon nitride' samples prepared from urea has already been reported by Martin et al, Angew Chem Int Ed 53 (2014) 9240. That publication reports a thorough examination of the chemical composition and structure of the material produced, as well as evaluation of the electronic structure from advanced ab initio calculations. Once more, this work is missing from the reference list.

Another key paper that is missing is the work by Schwarz et al (Adv Mater, 2017, 1703399) on mixed 2D/3D carbon nitride nanostructures grown from Cu-based solution, and containing Cu nanoparticles. Here also the Cu nanoparticles are characterised by XPS, and the same conclusion is reached — that the particles are either Cu(0) or Cu(I) in nature. The H₂ evolution by photo/electrocatalysis of these heterostructures is evaluated in that paper.

The manuscript can not be accepted in its current form. The authors need to complete their understanding of their materials and photocatalysis results and pay proper attention to previously published work, and use that to construct a more detailed and specific evaluation of their data

and the likely mechanisms involved. Those are critical aspects that will be important for future synthesis and processing of materials for large-scale applications.

Response

Thanks for the valuable suggestions. The description about ‘Holy Grail’ in the manuscript has been corrected.

The mentioned key recent papers (*Kessler et al., Nat. Rev. Mater.* 2 (2017) 17030; *Miller et al., Phys. Chem. Chem. Phys.* 19 (2017) 15613; *Martin et al., Angew. Chem. Int. Ed.* 53 (2014) 9240; *Schwarz et al., Adv. Mater.* (2017) 1703399) have been cited and consulted. Careful inspection of the characterizations of the photocatalyst was also conducted. The element content of the sample ‘g-C₃N₄’ and ‘Cu-0.5/g-C₃N₄’ from XPS analyses was listed in Table R3. The ratio of C to N from both the samples was about 3:4 rather than 2:3 of the Liebig’s melon. This result revealed that the surface composition of the photocatalysts was close to g-C₃N₄. However, considering that the thermal condensation of urea to obtain carbon nitrides would normally lead to incomplete polycondensation processes, it was hard to distinguish if it was strictly graphitic just from the XRD, FTIR, TEM and XPS analyses in our case. Thus, to be prudent and to avoid misleading expressions, we have corrected the designation of the photocatalyst from ‘graphitic carbon nitride (g-C₃N₄)’ to ‘polymeric carbon nitride (PCN)’.

Table R3. Element content of PCN and Cu-0.5/PCN from XPS analyses.

Atomic %	C	N	O	Cu
PCN	39.55	54.63	5.82	0
Cu-0.5/PCN	39.27	52.35	8.07	0.31

For photocatalytic methane conversion into ethanol, we provided more evidences for understanding this process and the possible mechanisms. The adsorption behavior of the materials and the stability of Cu-0.5/PCN were further studied. Meanwhile, the byproducts of photocatalytic methane conversion were analyzed and a methane-methanol-ethanol pathway was proposed. All the detailed description are displayed as follows (see also the response for comment 5, 6, and 7 from Reviewer #1).

To gain more insight about the influence of Cu introduction on the adsorption behavior, methane temperature programmed desorption (TPD) of PCN and Cu-0.5/PCN were conducted. As shown in Fig. R9, the physisorption of methane (around 100 °C) on PCN was stronger than Cu-0.5/PCN, agreeing with the unchanged intense signal of the *in situ* IR spectra on PCN. After introducing Cu into PCN, the peak around 250 °C demonstrated the chemisorption of methane,

agreeing with the progressively increasing signal of the *in situ* IR spectra on Cu-0.5/PCN. This result implied that PCN favored methane enrichment and Cu species coordinated into PCN played a key role in the C–H activation.

Fig. R9. Methane TPD of PCN and Cu-0.5/PCN.

The time course of the photocatalytic methane conversion over Cu-0.5/PCN was extended to 24 h (Fig. R10) and the XPS spectra of the sample after cycling tests (Fig. R11) were studied. In the C 1s XPS spectra, the peak at 284.5 eV became obvious, corresponding to coke deposition and agreeing with the ethanol production decay on Cu-0.5/PCN. The binding energy of all the peaks in the N 1s spectra was smaller in comparison to the fresh photocatalyst, which implied the increasing of the electron density of all kinds of N atoms. This result revealed that the disappeared hydrogen atoms were not only trapped by the edge N atoms but also trapped by the sp^3 -bonded N atoms ($C-NH_x$ and $N-(C)_3$). From the Cu 2p XPS spectra, the mixed-valence state remained unchanged after photocatalytic tests, indicating that the oxidized Cu species from H_2O_2 decomposition were reduced by the photo-induced electrons.

Fig. R10. Time course of photocatalytic methane-to-ethanol conversion over Cu-0.5/PCN for 24 h.

Fig. R11. C 1s (a), N 1s (b), and Cu 2p (c) XPS spectra of Cu-0.5/PCN before and after 24 h tests.

To gain more insight into the hypothetical mechanism, we attempted to quantify the photocatalytic products of methane conversion over Cu-0.5/PCN in the gas-solid static condition by offline detection. As shown in Table R4, liquid products contained methanol and ethanol while gas products contained H₂, CO and ethane. With methanol and ethane as byproducts, the mechanism was further refined as follows. The generation of ·OH and the cycle of Cu species (Eq. 18 to Eq. 22) were still valid. After hydrogen abstraction of methane by ·OH (Eq. 23), some of the generated methyl radical underwent radical coupling to produce ethane (Eq. 24) and the other reacted with water to produce methanol (Eq. 25). The generated ethane would also be attacked by ·OH to produce ethyl radical (Eq. 26), and the following ethanol was acquired from the reaction between ethyl radical and water (Eq. 27).

(26)

Table R4. Photocatalytic products of methane conversion over Cu-0.5/PCN ^a.

Liquid Product ($\mu\text{mol g}_{\text{cat}}^{-1} \text{h}^{-1}$)		Gas Product ($\mu\text{mol g}_{\text{cat}}^{-1} \text{h}^{-1}$)		
CH ₃ OH	CH ₃ CH ₂ OH	H ₂	CO	C ₂ H ₆
5.5	21.0	7.0	2.7	13.9

^a Gas-solid static condition: 20 mg of photocatalyst strewed in a glass dish surrounded by 25 mL of water, CH₄/N₂ atmosphere, 500 W Xe-lamp irradiating for 1h.

According to the radical mechanism, the generation of methanol in the liquid-solid dynamic condition should be more efficient like PCN because the content of methane was far above ethane. However, for Cu-X/PCN, the production rate of methanol was inferior to that of ethanol (Fig. R12). Given the poor solubility of alkane in water, there might be another mechanism dominating methanol conversion into ethanol for Cu modified PCN.

Fig. R12. Liquid products of methane conversion over PCN and Cu-X/PCN.

Synergistic active centers in Cu-C₃N₄ facilitating electrocatalytic CO₂ reduction to C₂ products was reported recently (*Jiao et al., J. Am. Chem. Soc. 139 (2017) 18093*), of which Cu species coordinated to the carbonous intermediates while the adjacent C atom in C₃N₄ coordinated to the oxygenous ones. Among the mentioned intermediates, hydroxymethyl group and methoxy group could be also derived from the interaction between methanol and $\cdot\text{OH}$. Hence, the dual active

center model of Cu species and the adjacent C atom in PCN might be applicable in our case as well. A series of experiments were carried out to validate the conjecture (Table R5). We started with introducing a small amount of methanol into the system for photocatalytic methane conversion (Entry 2 and 5). A significant increase of ethanol production on both PCN and Cu-0.5/PCN was achieved, but more methanol was consumed on PCN than that on Cu-0.5/PCN. Next, the experiments with methanol in the absence of methane revealed that more methanol was converted into ethanol with Cu modification, while on PCN it just decomposed, further confirming the role of methanol as a key intermediate (Entry 3 and 6). On the base of the results above, another hypothetic mechanism for methane conversion into ethanol through a methane-methanol-ethanol pathway was proposed. Hydrogen abstraction of the intermediate methanol by $\cdot\text{OH}$ generated hydroxymethyl and methoxy groups (Eq. 29 and Eq. 30). Hydroxymethyl or methyl groups binding on Cu species coupled with methoxy groups binding on the adjacent C atom in PCN to produce ethanol or ethyl radical, leaving an adsorbed O atom or hydroxyl group, which reacted with hydrogen atoms to form H_2O (Eq. 31 to Eq. 34). It was the synergy of the dual active center that brought about an improved ethanol production over Cu modified PCN.

(32)

Table R5. Experiments of methane conversion over PCN and Cu-0.5/PCN for the dual active center model^a.

Entry	Catalyst	Medium	Atmosphere	CH ₃ OH (μmol)	C ₂ H ₅ OH (μmol)
1	PCN	H ₂ O	CH ₄ /N ₂	0.39	0.11
2	PCN	7.5 μmol CH ₃ OH in H ₂ O	CH ₄ /N ₂	2.37	0.47
3	PCN	7.5 μmol CH ₃ OH in H ₂ O	N ₂	0.82	0.40
4	Cu-0.5/PCN	H ₂ O	CH ₄ /N ₂	0.47	2.12
5	Cu-0.5/PCN	7.5 μmol CH ₃ OH in H ₂ O	CH ₄ /N ₂	7.30	3.03
6	Cu-0.5/PCN	7.5 μmol CH ₃ OH in H ₂ O	N ₂	4.41	1.22

^a Liquid-solid dynamic condition: 20 mg of photocatalyst suspended in 25 mL of medium and kept stirring, 100 mL min⁻¹ of gas flow, 500 W Xe-lamp irradiating for 1 h.

Detailed revision please refer to **Appendix 2-1**, 4, 11, 14, 16, 17, 18, 19, 20, 28, 29, 30, 31, 34, 35, 36.

Reviewer #3

In this work, the authors report the design of Cu/g-C₃N₄ materials with tuned properties, for the partial photocatalytic oxidation of methane to ethanol. In fact, this is a very interesting and rather elusive topic, as these kinds of selective reactions under photocatalytic approaches tend to be difficult, considering the implicit need to balance the oxidizing species in order to avoid total oxidation to CO₂. In this sense, the proposed approach is quite promising, the analysis and experiment design are properly directed, and this approach seems to successfully overcome some of the intrinsic limitations of this reaction, so that it might constitute a reference for future concepts into selective photo-oxidation of methane. Despite this, some aspects should be better addressed and some points must be clarified.

1. Please re-write sentence in lines 45-46, page 3: 'which is rather low that no matter improvement...', as it doesn't seem very clear.

Response

Thanks for the suggestion. We have rephrased the sentence to avoid vague expression. Detailed revision please refer to **Appendix 2-5**.

2. Please explain lines 59-60 in the same page.

Response

Thanks for the suggestion. There are two key points about methane activation. The first one is methane activation by reactive species like ·OH. Generally, it leads to a thorough activation that the C–H bond of methane is broken and intermediate species such as methyl radical are formed. However, it is usually hard to control the following reaction of the free radical to obtain desired products. Confining methane and the intermediate species upon the surface of photocatalyst would be helpful. So, from our perspective, the second key point is methane activation through adsorption. The interaction between methane molecule and the surface of photocatalyst would induce subtle change to the perfect tetrahedral symmetry of methane, which could be construed as methane activation.

We have revised relative text in order to avoid vague expressions. Detailed revision please refer to **Appendix 2-6**.

3. Methane monooxygenase is first mentioned in the Introduction. I understand this is a particular enzymatic example that inspired the design of the catalytic structure proposed. It should be more clearly stated as it seems odd in the way it is included.

Response

Thanks for the valuable suggestion. The description of the active unit of methane monooxygenase and its connection with our design have been added. Detailed revision please refer to **Appendix 2-8**.

4. In XRD, Figure S1, although the background is relatively noisy, there are some zones in which broad signals seem to be present, such as at 23° and 34° 2theta, for the Cu-0.5/g-C₃N₄. Do these signals correspond to any other phase?

Response

Thanks for the suggestion. Careful inspection of the XRD patterns was carried out. We found that these signals were just noises.

5. Regarding g-C₃N₄ characterization, a more in-deep analysis should be performed, considering that urea thermal condensation for obtaining carbon nitrides, normally leads to incomplete polycondensation processes. Is there any remaining unreacted precursor or any influence of the CuCl₂ on the final structure?

Response

Thanks for the valuable suggestion. All the samples after thermal condensation were washed with water several times to remove the unreacted urea. The element content of g-C₃N₄ and Cu-0.5/g-C₃N₄ from XPS analyses (Table R6) revealed that the ratio of C to N was about 3:4. However, we found it hard to distinguish if it was strictly graphitic just from the XRD, FTIR, TEM and XPS analyses. Thus, to be prudent, we have changed the designation of the photocatalyst from graphitic carbon nitride (g-C₃N₄) to polymeric carbon nitride (PCN).

Table R6. Element content of g-C₃N₄ and Cu-0.5/g-C₃N₄ from XPS analyses.

Atomic %	C	N	O	Cu
g-C ₃ N ₄	39.55	54.63	5.82	0
Cu-0.5/g-C ₃ N ₄	39.27	52.35	8.07	0.31

After Cu modification, the intensity ratio of 13.0° (in-plane packing) to 27.4° (interfacial stacking) peaks in the XRD patterns gradually decreased from 0.37 to 0.18 by incorporating more Cu into g-C₃N₄, revealing that the interfacial stacking periodicity of g-C₃N₄ had been destructed.

Detailed revision please refer to **Appendix 2-10**, 11, 32, 36.

6. Was Cl analysed or detected by XPS?

Response

Thanks for the suggestion. Cl was not detected by XPS (Fig. R13). It was carried away in the form of HCl during the thermal condensation.

Fig. R13. Cl 2p XPS spectrum of Cu-0.5/g-C₃N₄.

7. Regarding the Mott-Schottky measurements, it is true that the band structure proposed seems to more or less agree with the expected band positions for the g-C₃N₄. However, it is important to consider that Mott-Schottky conditions might not be fully satisfied: for instance, M-S theory applies to flat surfaces (non-porous) and should be measured under high enough frequencies. In fact, there is a certain variation in the slopes with the frequency, despite the flat band potential seems to be independent from the frequency. I would suggest, at least, not to state ‘exact conduction band positions’, but to indicate it is just an approximation.

Response

Thanks for the valuable suggestion. We have corrected the relative description. Detailed revision please refer to **Appendix 2-12**.

8. In line 144, what do the authors mean by ‘photogenerated electron attack’?

Response

As shown by the following equation, the photogenerated electron attack meant that H₂O₂ was reduced directly by the photogenerated electron to produce ·OH. We have supplemented the following equation in the manuscript. Detailed revision please refer to **Appendix 2-13**.

9. The results suggest that H_2O_2 and further $\cdot OH$ generation on the $Cu-0.5/g-C_3N_4$ because of the presence of Cu . This effect might be interesting. Were any tests with the intermediate Cu compositions carried out, in order to see the intrinsic variation of both H_2O_2 and hydroxyl radicals, so that it can be properly correlated to the Cu content?

Response

Thanks for the suggestion. It is an insightful perspective to consider the influence of intermediate Cu compositions and the intrinsic variation of the reactive oxygen species. Unfortunately, the *in situ* characterizations could not be realized. In order to have insight into this important issue, the photocatalytic generation of H_2O_2 , $\cdot OH$ and the liquid products obtaining by offline detection were first summarized as shown in Fig. R14. Compared to $g-C_3N_4$, $Cu-0.5/g-C_3N_4$ generated less H_2O_2 , more $\cdot OH$ and more liquid products, implying that Cu introduction facilitated H_2O_2 decomposition into $\cdot OH$ which promoted methane conversion. Then, we recorded the Cu 2p XPS spectrum of $Cu-0.5/g-C_3N_4$ after cycling tests for 24 h (Fig. R15). The mixed-valence state remained unchanged after photocatalytic tests, indicating that the oxidized Cu species from H_2O_2 decomposition were reduced by the photo-induced electrons. This result also demonstrated the dynamic nature of the valence state of Cu species under irradiation. With the results above, we suppose that during photocatalytic methane conversion, the amount of both H_2O_2 and $\cdot OH$ increased in the initial stage then decreased to reach the equilibrium.

Fig. R14. The photocatalytic generation of H_2O_2 , $\cdot OH$ and the liquid products over $g-C_3N_4$ and $Cu-0.5/g-C_3N_4$ obtaining by offline detection.

Fig. R15. Cu 2p XPS spectra of Cu-0.5/g-C₃N₄ before and after 24 h tests.

10. In line 155-156 the authors mention the active sites for methane adsorption and activation. Both seem to be related to Cu species. Are these active sites associated to the same form of Cu?

Response

Thanks for the suggestion. To gain more insight about the influence of Cu introduction on the adsorption behavior, methane temperature programmed desorption (TPD) of g-C₃N₄ and Cu-0.5/g-C₃N₄ were conducted. As shown in Fig. R16, the physisorption of methane (around 100 °C) on g-C₃N₄ was stronger than Cu-0.5/g-C₃N₄, implying that g-C₃N₄ might benefit methane enrichment. After introducing Cu into g-C₃N₄, the peak around 250 °C demonstrated the chemisorption of methane, indicating that Cu species played a key role in the C–H activation of methane. Considering that the coordination between Cu species and the g-C₃N₄ was also important to methane activation and the dynamic nature of the Cu species under irradiation, we were not able to ascribe the property to a specific form of Cu (Cu⁰, Cu^I or Cu^{II}) solely.

Fig. R16. Methane TPD of g-C₃N₄ and Cu-0.5/g-C₃N₄.

11. After the IR measurements in gas-phase, the authors proposed an enhanced CH₄ adsorption on Cu sites. How does the balance between surface ·OH species and adsorbed CH₄ take place, if both are favored on Cu sites?

Response

Thanks for the suggestion. Considering the dynamic nature of the Cu species under irradiation, the coordination between Cu species and the g-C₃N₄ might also be dynamic, during which the Cu species would coordinate simultaneously with surface ·OH species and methane molecule.

Furthermore, we found that a dual active center model of Cu species and the adjacent C atom in g-C₃N₄ might be applicable in our case as well, of which the adjacent C atom showed strong binding to the reaction intermediates having oxygen as the connecting atom (e.g., *OCH₃, *O, *OH). Thus, the surface ·OH species generated from H₂O₂ decomposition might be transferred to the adjacent C atom site. With this model, a methane-methanol-ethanol pathway was also proposed to be responsible for photocatalytic methane conversion. Detailed analyses and a series of experiments to verify the dual active center model are described as follows (see also the response for comment 5, 6, and 7 from Reviewer #1).

First we attempted to quantify the photocatalytic products of methane conversion over Cu-0.5/g-C₃N₄ in the gas-solid static condition by offline detection. As shown in Table R7, liquid products contained methanol and ethanol while gas products contained H₂, CO and ethane. With methanol and ethane as byproducts, the mechanism was further refined as follows. The generation of ·OH and the cycle of Cu species (Eq. 35 to Eq. 39) were still valid. After hydrogen abstraction of methane by ·OH (Eq. 40), some of the generated methyl radical underwent radical coupling to produce ethane (Eq. 41) and the other reacted with water to produce methanol (Eq. 42). The

generated ethane would also be attacked by $\cdot\text{OH}$ to produce ethyl radical (Eq. 43), and the following ethanol was acquired from the reaction between ethyl radical and water (Eq. 44).

Table R7. Photocatalytic products of methane conversion over Cu-0.5/g-C₃N₄^a.

Liquid Product ($\mu\text{mol g}_{\text{cat}}^{-1} \text{h}^{-1}$)		Gas Product ($\mu\text{mol g}_{\text{cat}}^{-1} \text{h}^{-1}$)		
CH ₃ OH	CH ₃ CH ₂ OH	H ₂	CO	C ₂ H ₆
5.5	21.0	7.0	2.7	13.9

^a Gas-solid static condition: 20 mg of photocatalyst strewed in a glass dish surrounded by 25 mL of water, CH₄/N₂ atmosphere, 500 W Xe-lamp irradiating for 1h.

According to the radical mechanism, the generation of methanol in the liquid-solid dynamic condition should be more efficient like g-C₃N₄ because the content of methane was far above ethane. However, for Cu-X/g-C₃N₄, the production rate of methanol was inferior to that of ethanol (Fig. R17). Given the poor solubility of alkane in water, there might be another mechanism dominating methanol conversion into ethanol for Cu modified g-C₃N₄.

Fig. R17. Liquid products of methane conversion over g-C₃N₄ and Cu-X/g-C₃N₄.

Synergistic active centers in Cu-C₃N₄ facilitating electrocatalytic CO₂ reduction to C₂ products was reported recently (*Jiao et al., J. Am. Chem. Soc. 139 (2017) 18093*), of which Cu species coordinated to the carbonous intermediates while the adjacent C atom in C₃N₄ coordinated to the oxygenous ones. Among the mentioned intermediates, hydroxymethyl group and methoxy group could be also derived from the interaction between methanol and ·OH. Hence, the dual active center model of Cu species and the adjacent C atom in g-C₃N₄ might be applicable in our case as well. A series of experiments were carried out to validate the conjecture (Table R8). We started with introducing a small amount of methanol into the system for photocatalytic methane conversion (Entry 2 and 5). A significant increase of ethanol production on both g-C₃N₄ and Cu-0.5/g-C₃N₄ was achieved but more methanol was consumed on g-C₃N₄ than that on Cu-0.5/g-C₃N₄. Next, the experiments with methanol in the absence of methane revealed that more methanol was converted into ethanol with Cu modification, while on g-C₃N₄ it just decomposed, further confirming the role of methanol as a key intermediate (Entry 3 and 6). On the base of the results above, another hypothetic mechanism for methane conversion into ethanol through a methane-methanol-ethanol pathway was proposed. Hydrogen abstraction of the intermediate methanol by ·OH generated hydroxymethyl and methoxy groups (Eq. 46 and Eq. 47). Hydroxymethyl or methyl groups binding on Cu species coupled with methoxy groups binding on the adjacent C atom in g-C₃N₄ to produce ethanol or ethyl radical, leaving an adsorbed O atom or hydroxyl group, which reacted with hydrogen atoms to form H₂O (Eq. 48 to Eq. 51). It was the synergy of the dual active center that brought about an improved ethanol production over Cu modified g-C₃N₄.

(49)

Table R8. Experiments of methane conversion over g-C₃N₄ and Cu-0.5/g-C₃N₄ for the dual active center model^a.

Entry	Catalyst	Medium	Atmosphere	CH ₃ OH (μmol)	C ₂ H ₅ OH (μmol)
1	g-C ₃ N ₄	H ₂ O	CH ₄ /N ₂	0.39	0.11
2	g-C ₃ N ₄	7.5 μmol CH ₃ OH in H ₂ O	CH ₄ /N ₂	2.37	0.47
3	g-C ₃ N ₄	7.5 μmol CH ₃ OH in H ₂ O	N ₂	0.82	0.40
4	Cu-0.5/g-C ₃ N ₄	H ₂ O	CH ₄ /N ₂	0.47	2.12
5	Cu-0.5/g-C ₃ N ₄	7.5 μmol CH ₃ OH in H ₂ O	CH ₄ /N ₂	7.30	3.03
6	Cu-0.5/g-C ₃ N ₄	7.5 μmol CH ₃ OH in H ₂ O	N ₂	4.41	1.22

^a Liquid-solid dynamic condition: 20 mg of photocatalyst suspended in 25 mL of medium and kept stirring, 100 mL min⁻¹ of gas flow, 500 W Xe-lamp irradiating for 1 h.

Detailed revision please refer to **Appendix 2-16, 18, 19, 20, 28, 29, 30, 31.**

12. Regarding the blank mentioned tests, please detail how these blank tests were carried out: in absence of catalyst with methane; with catalyst in absence of methane; dark condition tests?

Response

Thanks for the suggestion. The details of the blank experiments are as follows.

Experiment with methane in absence of photocatalyst. Deionized water (25 mL) was added to the reactor and kept stirring. A mixture of methane (10 mL min⁻¹) and nitrogen (90 mL min⁻¹) was bubbled continuously through the water in the dark for 30 min and the illumination (500 W Xe-lamp) was turned on for 1 h. After that, the water was injected into the gas chromatographer to analyze the liquid composition.

Experiment with photocatalyst in absence of methane. A suspension of deionized water (25 mL) with the photocatalyst (20 mg) was added to the reactor and kept stirring. Nitrogen (100 mL min⁻¹) was bubbled continuously through the suspension in the dark for 30 min and the illumination (500 W Xe-lamp) was turned on for 1 h. After that, filtrate from the suspension was injected into the gas chromatographer to analyze the liquid composition.

Dark condition experiment. A suspension of deionized water (25 mL) with the photocatalyst (20 mg) was added to the reactor and kept stirring. A mixture of methane (10 mL min⁻¹) and nitrogen (90 mL min⁻¹) was bubbled continuously through the suspension in the dark for 90 min. After that, filtrate from the suspension was injected into the gas chromatographer to analyze the liquid composition.

13. Was any other possible product analyzed and/or obtained, such as CH₃OH?

Response

Thanks for the suggestion. We tried to analyze the photocatalytic products of methane conversion over Cu-0.5/g-C₃N₄ in the gas-solid static condition by offline detection. As shown in Table R7, liquid products contained methanol and ethanol while gas products contained H₂, CO and ethane.

14. In the experimental part, please indicate characteristic such as purity of the reactants.

Response

Thanks for the suggestion. All the chemicals involved were of analytical grade and used without further purification. We have complemented the purity of the reactants in the manuscript. Detailed revision please refer to **Appendix 2-23**.

15. Please indicate if the thermal condensation of the g-C₃N₄ was carried out under air or inert atmosphere.

Response

Thanks for the suggestion. The thermal condensation of g-C₃N₄ and Cu-X/g-C₃N₄ was carried out under air atmosphere. We have complemented the condition in the manuscript. Detailed revision please refer to **Appendix 2-24**.

16. Please explain better lines 236-237, as the volume of CuCl₂ solution does not seem to agree with the weight percentage of Cu/urea.

Response

Thanks for the suggestion. For the preparation of pristine g-C₃N₄, we received about 0.63 g material from the thermal condensation of 15 g urea. Thus we used the weight of g-C₃N₄ (0.63 g) to calculate the theoretical weight percentage of Cu. For the Cu-0.5/g-C₃N₄ sample, 0.5 mL of 0.1 mol L⁻¹ copper chloride contained 3.175 mg Cu, corresponding to about 0.5 wt%.

17. Please provide details on how the CH₄ adsorption tests were carried out.

Response

Thanks for the suggestion. The detail of *in situ* IR test is as follows. The photocatalyst (5 mg) was put and flattened in the infrared irradiation area (5 mm of diameter). After background subtraction, a gas mixture of 10% methane in nitrogen (30 mL min⁻¹) was introduced into the chamber and the IR spectra were recorded every 3 min.

18. Please state the irradiance used during the photocatalytic tests.

Response

Thanks for the suggestion. All the photocatalytic experiments were conducted using a 500 W Xe-lamp with an irradiation area of about 28 cm² (60 mm of spot diameter). We have complemented the illumination condition in the manuscript. Detailed revision please refer to **Appendix 2-26**.

19. Regarding the photocatalytic tests: the CH₄/N₂ mixture was continuously supplied to the system during all the test, but the ethanol analyzed from the liquid fraction corresponds to the accumulated alcohol produced. Is it correct? Was the gas phase analyzed in order to assess the possible generation of by-products or CO₂? Was the temperature constant during the test? Were the tests carried out several times for validating reproducibility?

Response

Thanks for the suggestion. It is correct that the gas mixture was continuously supplied to the system during the dynamic test and the ethanol analyzed from the liquid fraction corresponded to the accumulated alcohol. We tried to analyze the gas phase byproducts in the gas-solid static condition. H₂, CO and ethane were detected as gas phase byproducts while CO₂ was not detected. The temperature was not controlled during the test. The tests were carried out several times to ensure the reproducibility.

20. In the H₂ test, was the CH₄/N₂ mixture initially used for saturating the suspension but the gas flow was stopped before the photocatalytic test?

Response

Thanks for the suggestion. In the hydrogen production test, considering the poor solubility of methane, the photocatalyst was not suspended in water but strewed in a glass dish in the reactor.

The glass dish was surrounded by 25 mL of deionized water. A gas mixture of 10% methane in nitrogen was used for saturating the reactor. Then the gas flow was stopped and the reactor was sealed before the photocatalytic test.

21. As an additional comment, in some sections of the work, the authors mention a 'synergistic' effect. Albeit I understand it is associated to a better effect than the sum of the single elements (i.e., g-C₃N₄ and Cu), I would suggest to change this word, as the individual effect was not quantified in order to assess it is synergistic.

Response

Thanks for the valuable suggestion. We have corrected the relative description. Detailed revision please refer to **Appendix 2-2, 3, 9, 15, 21.**

Appendix 2

1. In the manuscript, all the phrases ‘graphitic carbon nitride (g-C₃N₄)’ were changed into ‘polymeric carbon nitride (PCN)’.
2. In Page 1, the sentence ‘A gentle path to generate ·OH and synergistic active site to adsorb and activate methane are vital for this process.’ was changed into ‘A gentle path to generate ·OH and active sites to adsorb and activate methane are vital for this process.’.
3. In Page 1, the sentences ‘These features synergistically avoided excess ·OH for overoxidation and facilitated methane conversion. Moreover, the mixed-valence Cu species were maintained during the photocatalytic cycle, which might be crucial to obtain C₂ products.’ were changed into ‘These features avoided excess ·OH for overoxidation and facilitated methane conversion. Moreover, a hypothetical mechanism through a methane-methanol-ethanol pathway was proposed, emphasizing the synergy of Cu species and the adjacent C atom in PCN for obtaining C₂ product.’.
4. In Page 2, the sentence ‘However, methane conversion to its alcohol derivatives is difficult to master and has rightfully emerged as one of the ‘Holy Grails of catalysis’⁴.’ was changed into ‘However, methane conversion to its alcohol derivatives is difficult to master and has rightfully emerged as the ‘Holy Grail’ of catalysis⁴.’.
5. In Page 3, the sentence ‘In these researches, the highest methanol production rate is 67.5 μmol g_{cat}⁻¹ h⁻¹ from WO₃ by using electron scavenger¹², which is rather low that no matter improvement of liquid fuel production or refinement of material design calls for thorough investigation.’ was changed into ‘In these researches, the highest methanol production rate is 67.5 μmol g_{cat}⁻¹ h⁻¹ from WO₃ by using electron scavenger¹², which is still far from satisfaction. Thus, refinement of material design and in-depth understanding of the mechanism are needed to improve the production of liquid fuel.’.
6. In Page 3, the sentence ‘In another sense, methane activation is the same critical as *in situ* production of the reactive oxygen species.’ was changed into ‘In another sense, methane activation through adsorption is the same critical as *in situ* production of the reactive oxygen species. The interaction between methane molecule and the surface of photocatalyst would

induce subtle change to the perfect tetrahedral symmetry of methane, which would be helpful for selective methane conversion.’.

7. In Page 4, the sentence ‘To achieve the objective, graphitic carbon nitride (g-C₃N₄) is a competent candidate.’ was changed into ‘To achieve the objective, polymeric carbon nitride (PCN) is a competent candidate, which has been investigated as one of the potential next-generation materials for energy conversion and catalysis application²⁰⁻²³.’.
8. In Page 4, the sentence ‘In the present work, we intended to introduce Cu species into the cavity of g-C₃N₄ inspired by its capability of H₂O₂ decomposition and the active sites of particulate methane monooxygenase in methanotrophic bacteria^{17, 24}.’ was changed into ‘Considering the mild conditions for methane conversion, the active site of enzyme offers excellent reference to search the potential components for introducing into PCN. In the methanotrophic bacteria, particulate methane monooxygenase is a critical metalloenzyme with a unique Cu cluster as the active site of methane hydroxylation²⁸. In the present work, we intended to introduce Cu species into the cavity of PCN inspired by its capability of H₂O₂ decomposition and the active site of particulate methane monooxygenase.’.
9. In Page 4, the sentences ‘These features avoided excess ·OH for deep mineralization and facilitated photocatalytic anaerobic methane conversion, synergistically leading to an ethanol generation rate of 106 μmol g_{cat}⁻¹ h⁻¹ under visible light. Moreover, maintaining the mixed valence states of Cu species through the photocatalytic cycle might be one of the key factors to obtain C₂ products.’ were changed into ‘These features avoided excess ·OH for deep mineralization, facilitating photocatalytic anaerobic methane conversion and generating ethanol as the main liquid product at a rate of 106 μmol g_{cat}⁻¹ h⁻¹ under visible light. Moreover, the synergy of Cu species and the adjacent C atom in PCN played a key role to obtain ethanol through a methane-methanol-ethanol pathway.’.
10. In Page 5, the sentences ‘Cu-X/g-C₃N₄ held similar patterns to that of pristine g-C₃N₄, which indicated that the general structure of g-C₃N₄ was preserved with Cu modification (Supplementary Fig. 1). In Fourier transform infrared (FTIR) spectra of g-C₃N₄ and Cu-0.5/g-C₃N₄ (Supplementary Fig. 2), no distinct changes occurred after Cu introduction, confirming that modification of g-C₃N₄ exerted negligible influence on its basic structure.’

were changed into ‘Cu-X/PCN held similar patterns to that of PCN without additional peaks. However, the intensity ratio of 13.0° (in-plane packing) to 27.4° (interfacial stacking) peaks gradually decreased from 0.37 to 0.18 by incorporating more Cu into PCN, revealing that the interfacial stacking periodicity of PCN had been destructed (Supplementary Fig. 1)^{25,26}. In Fourier transform infrared (FTIR) spectra of PCN and Cu-0.5/PCN (Supplementary Fig. 2), no distinct changes of the band positions occurred after Cu introduction, confirming that modification of PCN exerted a negligible influence on its basic structure.’.

11. In Page 6, new sentences ‘The element content of PCN and Cu-0.5/PCN was also analyzed (Supplementary Table 1). No residual Cl was detected, which might be carried out in the form of hydrogen chloride during thermal condensation.’ were added.
12. In Page 6, the sentence ‘Mott-Schottky plots (Fig. 2c) helped to estimate the exact conduction band positions, revealing band structure alignments (Fig. 2d) together with the bandgap energy assessed from diffuse reflectance spectroscopy (DRS) data.’ was changed into ‘Mott-Schottky plots (Fig. 2c) helped to estimate the approximate conduction band positions, revealing band structure alignments (Fig. 2d) together with the bandgap energy assessed from diffuse reflectance spectroscopy (DRS) data.’.
13. In Page 7, the sentence ‘In contrast, a more intense fluorescent signal of Cu-0.5/g-C₃N₄ implied that Cu modification accelerated the decomposition of H₂O₂ and produced more ·OH, while the weak signal on pristine g-C₃N₄ was attributed to the inefficient decomposition by photolysis or photogenerated electron attack¹⁷.’ was changed into ‘In contrast, a more intense fluorescent signal of Cu-0.5/PCN implied that Cu modification accelerated the decomposition of H₂O₂ and produced more ·OH, while the weak signal on PCN was attributed to the inefficient decomposition by photolysis or photogenerated electron attack (H₂O₂ + e⁻ → ·OH + OH⁻)¹⁷.’.
14. In Page 8, new sentences ‘Methane temperature programmed desorption (TPD) of PCN and Cu-0.5/PCN was also investigated to gain more insight into methane adsorption and activation.

As shown in Fig. 3d, the physisorption of methane (around 100 °C) on PCN was stronger than Cu-0.5/PCN, agreeing with the unchanged signal of the *in situ* IR spectra on PCN. After introducing Cu into PCN, the peak around 250 °C demonstrated the chemisorption of methane, agreeing with the progressively increasing signal of the *in situ* IR spectra on Cu-0.5/PCN³⁵. This result, together with the *in situ* IR spectra, implied that PCN favored methane enrichment and Cu species coordinated into PCN played a key role in the C–H activation.’ were added.

15. In Page 8, the sentence ‘To validate the synergistic effect of our design, photocatalytic anaerobic methane conversion tests were performed at room temperature and atmospheric pressure, generating ethanol as liquid product.’ was changed into ‘To validate the feasibility of our design, a series of experiments in liquid-solid dynamic condition for photocatalytic anaerobic methane conversion were performed at room temperature and atmospheric pressure while the accumulated liquid products were analyzed by gas chromatographer.’.

16. In Page 8, the sentences ‘As depicted in Fig. 3d, the productivity over pristine g-C₃N₄ was 5.5 $\mu\text{mol g}_{\text{cat}}^{-1} \text{h}^{-1}$, which seemed inconspicuous due to the absence of active sites for methane activation and *in situ* H₂O₂ decomposition. A significant increase of ethanol yield was achieved over Cu-X/g-C₃N₄, among which Cu-0.5/g-C₃N₄ reached the highest ethanol production rate of 106 $\mu\text{mol g}_{\text{cat}}^{-1} \text{h}^{-1}$ and remained stable over at least 5 cycles of testing (Supplementary Fig. 6). Furthermore, trace amount of H₂ (7 $\mu\text{mol g}_{\text{cat}}^{-1} \text{h}^{-1}$) was obtained as byproduct in a gas-solid static experiment (Fig. 3e).’ were changed into ‘As depicted in Fig. 3e, the liquid products contained methanol and ethanol. The variation of methanol productivity over all the samples was subtle. However, a significant increase of ethanol yield was achieved over Cu-X/PCN, among which Cu-0.5/PCN reached the highest ethanol production rate of 106 $\mu\text{mol g}_{\text{cat}}^{-1} \text{h}^{-1}$. The time course of the photocatalytic methane-to-ethanol conversion over Cu-0.5/PCN was carried out (Supplementary Fig. 6), indicating that the ethanol production rate decayed slightly over 24 h of testing. The XPS spectra of the sample after cycling tests were also studied. From the Cu 2p XPS spectra (Supplementary Fig. 7a), the mixed-valence state remained unchanged after photocatalytic tests, implying that the oxidized Cu species from H₂O₂ decomposition were reduced by the photogenerated electrons. The peak at 284.5 eV in the C 1s XPS spectra became obvious, corresponding to coke deposition and agreeing with the ethanol production decay on

Cu-0.5/PCN (Supplementary Fig. 7b). Furthermore, the photocatalytic gas byproducts of methane conversion over Cu-0.5/PCN containing H₂, CO and ethane were detected in the gas-solid static condition (Table 1).’.

17. In Page 8, the sentence ‘However, the H₂ evolution rate was rather low and nonstoichiometric to that of ethanol (21 μmol g_{cat}⁻¹ h⁻¹), we ascribed it to the strong coordination ability of edge N atoms on Cu-0.5/g-C₃N₄ to trap the hydrogen atoms³¹.’ was changed into ‘However, the H₂ evolution rate was rather low and nonstoichiometric to that of ethanol, we ascribed the disappeared H₂ to the strong coordination ability of the N atoms on Cu-0.5/PCN to trap the hydrogen atoms^{26,36}, which was confirmed in the N 1s XPS spectra (Supplementary Fig. 7c) that the electron density of all kinds of N atoms became bigger in comparison to the fresh photocatalyst.’.

18. In Page 9, the sentences ‘This initiation step activated the adsorbed methane and also consumed the generated ·OH right away, which could effectively avoid complete mineralization by excess ·OH. Subsequently, the coupling of two methyl radicals gave rise to ethyl radical, while the following ethanol was acquired from the reaction between ethyl radical and H₂O (equation (5) and (6)). That is to say, the core of photocatalytic methane conversion lies in controlling the generation of reactive oxygen species and activating methane. The appealing band structure alignments of g-C₃N₄ holds one key to obtain H₂O₂ through H₂O oxidation and reduce Cu species to accomplish photocatalytic cycle, while mixed-valence Cu species hold another to activate methane and decompose H₂O₂ into ·OH. These essentials synergistically facilitate methane conversion and mitigate the negative effect of excess ·OH, leading to an enhanced efficiency of photocatalytic methane conversion. Moreover, it is reported that the mixed valence states of Cu species were critical in electrocatalytic CO₂ reduction to obtain C₂ products^{34,35}. Thus, in our work, maintaining the mixed valence states of Cu species through the photocatalytic cycle might be one of the key factors to produce ethanol.’ were changed into ‘This initiation step activated the adsorbed methane and also consumed some of the generated ·OH, which could effectively avoid complete mineralization by excess ·OH. Subsequently, some of the generated methyl radical

underwent radical coupling to produce ethane (Eq. 7) and the other reacted with H₂O to produce methanol (Eq. 8). The ethane would also be attacked by ·OH to produce ethyl radical (Eq. 9), while the following ethanol was acquired from the reaction between ethyl radical and H₂O (Eq. 10).’.

19. In Page 10, the equations were changed into:

(7)

20. In Page 10, new paragraphs and new equations were added as follows.

According to the radical mechanism above, the generation of methanol in the liquid-solid dynamic condition should be more efficient like PCN (Fig. 3e) because the content of methane was far above ethane. However, for Cu-X/PCN, the production rate of methanol was inferior to that of ethanol. Given the poor solubility of alkane in H₂O, there might be another mechanism dominating methanol conversion into ethanol for Cu modified PCN.

In a previous report, Jiao *et al.* presented a synergistic effect in Cu-C₃N₄ facilitating electrocatalytic CO₂ reduction to C₂ products²⁶, of which Cu species coordinated to the carbonous intermediates while the adjacent C atom in C₃N₄ coordinated to the oxygenous ones. Among the mentioned intermediates, hydroxymethyl group and methoxy group could be also derived from the interaction between methanol and ·OH³⁹. Hence, the dual active center model of Cu species and the adjacent C atom in PCN might be applicable in our case as well. A series of experiments were carried out to validate the conjecture (Table 2). We started with introducing a small amount of methanol into the system for photocatalytic methane

conversion (Entry 2 and 5). A significant increase of ethanol production on both PCN and Cu-0.5/PCN was achieved but more methanol was consumed on PCN than that on Cu-0.5/PCN. Then, experiments with methanol in the absence of methane revealed that more methanol was converted into ethanol with Cu modification, while on PCN it just decomposed, further confirming the role of methanol as a key intermediate (Entry 3 and 6). On the base of the results above, another hypothetical mechanism for methane conversion into ethanol through a methane-methanol-ethanol pathway was proposed (Fig. 4). Hydrogen abstraction of the intermediate methanol by $\cdot\text{OH}$ generated hydroxymethyl and methoxy groups (Eq. 12 and Eq. 13). Hydroxymethyl or methyl groups binding on Cu species coupled with methoxy groups binding on the adjacent C atom in PCN to produce ethanol or ethyl radical, leaving an adsorbed O atom or hydroxyl group, which reacted with hydrogen atoms to form H_2O (Eq. 14 to Eq. 17).

(12)

That is to say, the core of photocatalytic methane conversion lies in controlling the generation of reactive oxygen species and activating methane. The appealing band structure alignments of PCN holds one key to obtain H_2O_2 through H_2O oxidation and reduce Cu species to accomplish photocatalytic cycle, while mixed-valence Cu species hold another to activate methane and decompose H_2O_2 into $\cdot\text{OH}$. These essentials facilitate methane conversion and largely mitigate the negative effect of excess $\cdot\text{OH}$, leading to an enhanced efficiency of photocatalytic methane conversion. For producing C_2 product, the synergy of Cu species and the adjacent C atom in PCN provides key contributions. It is also reported that the mixed-valence Cu species were critical in electrocatalytic CO_2 reduction to obtain C_2 products^{40, 41}. Thus, in our work, maintaining the mixed valence states of Cu species through the photocatalytic cycle might also be a key factor to obtain ethanol.

21. In Page 10, the sentence ‘This ingenious material design provided active sites for *in situ* generation of $\cdot\text{OH}$ as well as methane adsorption and activation, synergistically led to

enhanced photocatalytic anaerobic methane conversion and avoided complete mineralization.’ was changed into ‘This ingenious material design provided active sites for *in situ* generation of ·OH as well as methane adsorption and activation, led to enhanced photocatalytic anaerobic methane conversion and avoided complete mineralization.’.

22. In Page 11, a new sentence ‘Furthermore, the synergy between Cu species and the adjacent C atom in PCN played a key role to obtain C₂ product through a methane-methanol-ethanol pathway.’ was added.
23. In Page 11, a new sentence ‘All the chemicals involved were of analytical grade and used without further purification.’ was added.
24. In Page 11, the sentence ‘Briefly, 15 g of urea was calcined at 550 °C in muffle furnace for 4 h at a ramp rate of 10 °C min⁻¹.’ was changed into ‘Briefly, 15 g of urea was calcined at 550 °C in muffle furnace for 4 h under air atmosphere at a ramp rate of 10 °C min⁻¹.’.
25. In Page 12, a new sentence ‘Methane temperature programmed desorption (TPD) was performed on an AutoChem II 2920 chemisorption analyzer.’ was added.
26. In Page 12, the sentence ‘For the photocatalytic experiments, a customized photochemical reactor was used with a 500 W Xe-lamp.’ was changed into ‘For all the photocatalytic experiments, a customized photochemical reactor was used with a 500 W Xe-lamp (60 mm of spot diameter).’.
27. In Page 14, the sentence ‘After illumination for 2 h, the amount of hydrogen was determined by a Techcomp GC7890II gas chromatographer with a Molecular Sieve 5A 80/100 Mesh column and a thermal conductivity detector, while the liquid composition was examined by Techcomp GC7900 gas chromatographer.’ was changed into ‘After illumination for 2 h, the amount of hydrogen was determined by a Techcomp GC7890II gas chromatographer with a Molecular Sieve 5A 80/100 Mesh column and a thermal conductivity detector. Other gas phase products and the liquid composition were analyzed by a Techcomp GC7900 gas chromatographer with TDX-01, TM-Al₂O₃/S and SE-54 columns and flame ionization detectors.’.

28. In Page 14, new papers were added in the bibliography:

20. Kessler F. K., *et al.* Functional carbon nitride materials - design strategies for electrochemical devices. *Nat. Rev. Mater.* **2**, 17030 (2017).

21. Miller T. S., *et al.* Carbon nitrides: synthesis and characterization of a new class of functional materials. *Phys. Chem. Chem. Phys.* **19**, 15613-15638 (2017).

22. Martin D. J., *et al.* Highly efficient photocatalytic H₂ evolution from water using visible light and structure-controlled graphitic carbon nitride. *Angew. Chem. Int. Ed.* **53**, 9240-9245 (2014).

23. Schwarz D., *et al.* Twinned growth of metal-free, triazine-based photocatalyst films as mixed-dimensional (2D/3D) van der Waals heterostructures. *Adv. Mater.* **29**, 1703399 (2017).

35. Tabata T., Kokitsu M. & Okada O. Adsorption properties of oxygen and methane on Ga-ZSM-5; the origin of the selectivity of NO_x reduction using methane. *Catal. Lett.* **25**, 393-400 (1994).

39. Feng J., Aki S. N. V. K., Chateaneuf J. E. & Brennecke J. F. Abstraction of hydrogen from methanol by hydroxyl radical in subcritical and supercritical water. *J. Phys. Chem. A* **107**, 11043-11048 (2003).

29. In Page 21, Fig. 3 was replaced with the following graphic:

Fig. 3. Photocatalytic performance. (a) Photocatalytic anaerobic H_2O_2 production over PCN and Cu-0.5/PCN; (b) Fluorescent spectra of 2-hydroxyterephthalic acid for hydroxyl radical detection over PCN and Cu-0.5/PCN; (c) *In situ* IR spectra of methane adsorption on Cu-0.5/PCN; (d) Methane TPD of PCN and Cu-0.5/PCN; (e) Liquid products of methane conversion over PCN and Cu-X/PCN; (f) Photocatalytic methane conversion over Cu-0.5/PCN with or without O_2 .

Table 1. Photocatalytic products of methane conversion over Cu-0.5/PCN ^a.

Liquid Product ($\mu\text{mol g}_{\text{cat}}^{-1} \text{h}^{-1}$)		Gas Product ($\mu\text{mol g}_{\text{cat}}^{-1} \text{h}^{-1}$)		
CH ₃ OH	CH ₃ CH ₂ OH	H ₂	CO	C ₂ H ₆
5.5	21.0	7.0	2.7	13.9

^a Gas-solid static condition: 20 mg of photocatalyst strewed in a glass dish surrounded by 25 mL of water, CH₄/N₂ atmosphere, 500 W Xe-lamp irradiating for 1h.

Table 2. Experiments of methane conversion over PCN and Cu-0.5/PCN for the dual active center model^a.

Entry	Catalyst	Medium	Atmosphere	CH ₃ OH (μmol)	C ₂ H ₅ OH (μmol)
1	PCN	H ₂ O	CH ₄ /N ₂	0.39	0.11
2	PCN	7.5 μmol CH ₃ OH in H ₂ O	CH ₄ /N ₂	2.37	0.47
3	PCN	7.5 μmol CH ₃ OH in H ₂ O	N ₂	0.82	0.40
4	Cu-0.5/PCN	H ₂ O	CH ₄ /N ₂	0.47	2.12
5	Cu-0.5/PCN	7.5 μmol CH ₃ OH in H ₂ O	CH ₄ /N ₂	7.30	3.03
6	Cu-0.5/PCN	7.5 μmol CH ₃ OH in H ₂ O	N ₂	4.41	1.22

^a Liquid-solid dynamic condition: 20 mg of photocatalyst suspended in 25 mL of medium and kept stirring, 100 mL min⁻¹ of gas flow, 500 W Xe-lamp irradiating for 1 h.

31. In Page 23, Fig. 4 was replaced with the following graphic:

[Redacted]

Fig. 4. The hypothetic mechanism for photocatalytic anaerobic methane conversion over Cu-0.5/PCN.

32. In Supplementary Information (Page 2), Supplementary Figure 1 was replaced with the following graphic:

Supplementary Figure 1. XRD patterns of PCN and Cu-X/PCN. The interfacial stacking periodicity of PCN had been destructed with Cu introduction.

33. In Supplementary Information (Page 2), the caption of Supplementary Figure 2 was changed into ‘**Supplementary Figure 2. FTIR spectra of PCN and Cu-0.5/PCN.** No distinct changes of the band positions occurred after Cu introduction, confirming that modification of PCN exerted a negligible influence on its basic structure.’.

34. In Supplementary Information (Page 5), Supplementary Figure 6 was replaced with the following graphic:

Supplementary Figure 6. Time course of photocatalytic methane-to-ethanol conversion over Cu-0.5/PCN for 24 h. Ethanol production decayed slightly over 24 h of testing.

35. In Supplementary Information (Page 6), new graphics were added.

Supplementary Figure 7. Cu 2p (a), C 1s (b), and N 1s (c) XPS spectra of Cu-0.5/PCN before and after 24 h tests. From the Cu 2p XPS spectra, the mixed-valence state remained unchanged after photocatalytic tests, indicating that the oxidized Cu species from H₂O₂ decomposition were reduced by the photo-induced electrons. In the C 1s XPS spectra, the peak at 284.5 eV became obvious, corresponding to coke deposition and agreeing with the ethanol production decay on Cu-0.5/PCN. The binding energy of all the peaks in the N 1s spectra was smaller in comparison to the fresh photocatalyst, which implied the increasing of the electron density of all kinds of N atoms.

Supplementary Figure 8. Photocatalytic hydrogen evolution over Cu-0.5/PCN with and without methane. No H₂ was detected in the methane-free experiment, excluding H₂ evolution from photocatalytic H₂O splitting.

36. In Supplementary Information (Page 6), a new table was added.

Supplementary Table 1. Element content of PCN and Cu-0.5/PCN from XPS analyses.

Atomic %	C	N	O	Cu
PCN	39.55	54.63	5.82	0
Cu-0.5/PCN	39.27	52.35	8.07	0.31

REVIEWERS' COMMENTS:

Reviewer #1 (Remarks to the Author):

The authors adequately replied to the comments provided from the reviewer, and well revised the manuscript with the additional data. Thus, this manuscript is now recommendable for the publication in this journal.

Hisao Yoshida

Reviewer #2 (Remarks to the Author):

The authors have responded in detail to the questions raised by the reviewers.

It is still not clear exactly where the Cu is located within the material, what it is bound to, or even what its oxidation state truly is. The Cu XPS spectrum shown in SI is of very poor quality (a higher resolution version will need to be uploaded to be a useful addition to the paper: also the eV scale is cut off at the bottom). The Cu 2p data should be extended to examine the range up to 970 eV, to observe any contribution from the higher energy satellite contribution for Cu²⁺ species. As it stands, the XPS data seem to indicate only CuO and Cu⁺ species, with perhaps a slight increase in intensity in the region expected for the first Cu²⁺ satellite in the "after" spectrum. However the EPR data show the presence of the 2+ state. The authors have discussed this, but the actual incorporation of Cu into the material remains a mystery. All that the authors could do at this stage would be to investigate further using DFT calculations.

The O1s signal for both samples (with and without Cu) show that this element is incorporated in the structure, and the signal changes in shape when Cu is present. Why do the authors not conclude that Cu is coordinated to the "impurity" O atoms, rather than C or N sites? If they check the thermodynamic stability of Cu-O vs Cu-C/Cu-N bonds they might find that this is a more likely solution.

The fact that the material changes its UV-vis absorption profile as well as its methane oxidation activity following Cu incorporation does show that this simple chemical functionalization is beneficial, for this important reaction.

I would suggest the the paper can be published: it will certainly attract interest from a wide audience, even though the details are not all sorted out.

Reviewer #3 (Remarks to the Author):

As stated in the previous assessment and mentioned by the authors in the Introduction section, the topic and approach used in this work is of significant interest from the environmental point of view and for energy storage applications.

The authors have addressed a proper evaluation of the different comments posed by the reviewers, including some additional tests and a more thorough revision on the pertinent literature, not only related to the photocatalytic reaction, but also to the material itself.

Although full understanding on the mechanisms involved is not achieved, I consider that given the complexity of the process, the authors have carried out a good approximation for explaining and justifying the implicit reactions and the final results, by using standard characterization and analytical techniques. Probably, more specific techniques would be necessary in order to further elucidate these processes, but it would escape from the objective of the present work, especially in a short communication format. Moreover, the authors have also included the Experimental information requested, necessary for validation, reproducibility and future works.

I would recommend a more exhaustive English and figure edition. For instance, in Figure 2, it would be helpful using a different set of colors.

Appendix 1

Reviewer #1

The authors adequately replied to the comments provided from the reviewer, and well revised the manuscript with the additional data. Thus, this manuscript is now recommendable for the publication in this journal.

Hisao Yoshida

Response

Thanks. Your recommendation is highly appreciated.

Reviewer #2

The authors have responded in detail to the questions raised by the reviewers.

It is still not clear exactly where the Cu is located within the material, what it is bound to, or even what its oxidation state truly is. The Cu XPS spectrum shown in SI is of very poor quality (a higher resolution version will need to be uploaded to be a useful addition to the paper: also the eV scale is cut off at the bottom). The Cu 2p data should be extended to examine the range up to 970 eV, to observe any contribution from the higher energy satellite contribution for Cu²⁺ species. As it stands, the XPS data seem to indicate only Cu⁰ and Cu⁺ species, with perhaps a slight increase in intensity in the region expected for the first Cu²⁺ satellite in the "after" spectrum. However the EPR data show the presence of the 2+ state. The authors have discussed this, but the actual incorporation of Cu into the material remains a mystery. All that the authors could do at this stage would be to investigate further using DFT calculations.

The O 1s signal for both samples (with and without Cu) show that this element is incorporated in the structure, and the signal changes in shape when Cu is present. Why do the authors not conclude that Cu is coordinated to the "impurity" O atoms, rather than C or N sites? If they check the thermodynamic stability of Cu-O vs Cu-C/Cu-N bonds they might find that this is a more likely solution.

The fact that the material changes its UV-vis absorption profile as well as its methane oxidation activity following Cu incorporation does show that this simple chemical functionalization is beneficial, for this important reaction.

I would suggest the paper can be published: it will certainly attract interest from a wide audience, even though the details are not all sorted out.

Response

Thanks for your valuable suggestions and positive recommendation. It is an insightful perspective to investigate the material using DFT calculations. Unfortunately, we are unable to achieve the calculations. In order to have insight into the valence state and bonding situation of the Cu species, careful inspection of the XPS analyses was carried out.

The poor quality of the Cu 2p spectra and the Cu LMM spectrum was due to the low Cu content (Table R1). As shown in Fig. R1, the range of the Cu 2p spectra was extended to 970 eV. The satellite peaks corresponding to Cu^{II} were all submerged in the background before and after the cycling test. Thus, the Cu species were of Cu⁰ or Cu^I from the XPS data. However, under irradiation, the Cu species were oxidized during H₂O₂ decomposition then reduced by the photo-induced electrons. This dynamic nature made it hard to identify the exact oxidation state. Therefore, we described the oxidation state of the Cu species as ‘mixed-valance state’.

Table R1. Element content of PCN and Cu-0.5/PCN from XPS analyses.

Atomic %	C	N	O	Cu
PCN	39.55	54.63	5.82	0
Cu-0.5/PCN	39.27	52.35	8.07	0.31

Fig. R1. Cu 2p XPS spectra of Cu-0.5/PCN before and after the cycling test.

For the O 1s spectra (Fig. R2), the peaks of 532.7 and 531.5 eV could be assigned to $\text{H}_2\text{O}_{\text{ads}}$ and C=O respectively. The signal changed in shape because of the different O content between PCN and Cu-0.5/PCN (Table R1). In most cases, Cu–O bond was thermodynamically more stable than Cu–C or Cu–N bonds. However, after summarizing the binding energy of $\text{Cu}^{\text{I}}\text{–O}$ and $\text{Cu}^{\text{II}}\text{–O}$ bonds from the NIST XPS database (<http://dx.doi.org/10.18434/T4T88K>) and many articles (*He et al., J. Am. Chem. Soc.* 140 (2018) 1824; *Yang et al., Appl. Catal. B* 170 (2015) 225; *Martin et al., J. Phys. Chem. C* 117 (2013) 4421 etc.), we found them locating at around 530 eV ($\text{Cu}^{\text{I}}\text{–O}$) or 529 eV ($\text{Cu}^{\text{II}}\text{–O}$), which were obviously not included in the O 1s spectrum of Cu-0.5/PCN. Taken into account that no shift of the peaks occurred in the C 1s spectra, while one of the peaks in the N 1s spectra raised to higher binding energy after Cu modification, we came to a cautious conclusion that the Cu species were coordinated to the N atoms.

Fig. R2. O 1s XPS spectra of PCN and Cu-0.5/PCN.

Detailed revision please refer to **Appendix 2-9**.

Reviewer #3

As stated in the previous assessment and mentioned by the authors in the Introduction section, the topic and approach used in this work is of significant interest from the environmental point of view and for energy storage applications.

The authors have addressed a proper evaluation of the different comments posed by the reviewers, including some additional tests and a more thorough revision on the pertinent literature, not only related to the photocatalytic reaction, but also to the material itself.

Although full understanding on the mechanisms involved is not achieved, I consider that given the complexity of the process, the authors have carried out a good approximation for explaining and justifying the implicit reactions and the final results, by using standard characterization and analytical techniques. Probably, more specific techniques would be necessary in order to further elucidate these processes, but it would escape from the objective of the present work, especially in a short communication format. Moreover, the authors have also included the Experimental information requested, necessary for validation, reproducibility and future works.

I would recommend a more exhaustive English and figure edition. For instance, in Figure 2, it would be helpful using a different set of colors.

Response

Thanks for your valuable suggestions and positive recommendation. We have checked the manuscript again and corrected the improper or false description. Figure edition has also been conducted using a different set of colors. Detailed revision please refer to **Appendix 2-2**, 3, 4, 5, 6, 7, 8.

Appendix 2

1. In Page 1, lines 8-9, the sentence 'A gentle path to generate ·OH and active sites to adsorb and activate methane are vital for this process.' was deleted to satisfy the word limit.
2. In Page 2, line 37, 'by taking advantage of solar energy' was changed into 'by virtue of photoexcitation'.
3. In Page 4, line 81, 'introducing into PCN.' was changed into 'PCN modification.'.
4. In Page 7, lines 139-140, 'the bandgap energy assessed from diffuse reflectance spectroscopy (DRS) data.' was changed into 'the optical bandgap assessed from the data of diffuse reflectance spectroscopy (DRS).'.
5. In Page 8, line 152, 'of PCN' was changed into 'over PCN'.
6. In Page 10, line 205, 'ethanol' was changed into 'alcohols'.
7. In Page 10, line 209, 'bigger' was changed into 'higher'.
8. In the manuscript, the figures were edited using a different set of colors.
9. The range of the Cu 2p XPS spectra in Figure 1 and Supplementary Figure 7 was extended to 970 eV.